# A likely role for stratification in long-term changes of the global ocean tides
Lana Opel [1] ✉, Michael Schindelegger [1] & Richard D. Ray [2]

Stratification—that is, the vertical change in seawater density—exerts a subtle control on the energetics and thus the surface elevation of barotropic (depth independent) flows in the ocean. Changes in stratification therefore provide a plausible pathway to explain some of the puzzling trends in ocean tides evident in tide gauge and, more recently, satellite altimetry data. Using a three-dimensional global ocean model, we estimate that strengthening of stratification between 1993 and 2020 caused open-ocean trends of order 0.1 mm yr$^{-1}$ in the barotropic $M_2$ tide, similar in structure and magnitude to long-term $M_2$ changes deduced from satellite altimetry. Amplitude trends are predominantly negative, implying enhanced energy transfer to internal tides since the 1990s. Effects of stratification are also a relevant forcing of contemporary $M_2$ trends at the coast, where they may modulate or even overprint the tidal response to sea level rise. Salient examples for such significant near-shore influence of stratification (≥ 95% confidence) include the Northwest Australian Shelf (− 0.5 mm yr$^{-1}$) and the coasts of western North America (− 0.1 mm yr$^{-1}$), commensurate with observed $M_2$ amplitude trends at tide gauges.

Analyses of coastal tide gauge measurements spanning the past ~30–100 years have shown that ocean tides undergo unexpected changes on inter-annual, decadal, and secular time scales[1–9]. The changes themselves are rather subtle—typically ~1–3 cm in amplitude over a century—but large enough to rule out a connection to variations in the astronomical forcing[5]. At the local scale and in special regional settings[10], tidal properties may easily be affected by alterations of geometry, caused by, e.g., coastal engineering measures, wetland loss, or harbor dredging if we extend our scope slightly more inland[11]. However, most regional assessments (see references above) point to a certain degree of spatial coherence in the patterns of tidal change, likely bearing on the role of broader natural, rather than local anthropogenic factors. Satellite radar altimetry, now providing 30 years' worth of data, has begun to add to the picture drawn by tide gauges at the coast; recent analysis by Bij de Vaate et al.[12] suggests negative $M_2$ amplitude trends of ~− 0.3 mm yr$^{-1}$ across the subpolar North Atlantic and in the Gulf of Alaska.

Knowledge of what causes these changes can lay the groundwork for a better depiction of non-stationary tidal effects in projections of extreme sea levels[13] and nuisance flooding[14], and for the purpose of de-aliasing satellite altimetry and gravimetry observations[15,16]. On the subject of trends, modeling work has so far focused on the tidal response to contemporary sea level rise and isostatic crustal motion[17–20]. This is indeed a reasonable link to explore in very shallow environments (e.g., estuaries and shallow shelf seas), where water depth changes strongly project onto the bed friction term in the momentum equations. However, relative sea level rise can only explain fractions (≲20%) of the observed trends at most locations open to the sea and has little impact on $M_2$ at the basin scale[19].

Could changes in ocean stratification be a more relevant forcing factor? Analytical treatments[21] and case studies of interannual tidal variability[7,22] suggest that this question is well worth pursuing. In particular, as the upper-ocean warming signal strengthens stratification[23,24] (see also Fig. 1), barotropic tidal dynamics may be subject to two effects. First, increased vertical density contrast allows for enhanced generation and propagation of internal tides, initiated when depth-averaged tidal currents meet steep underwater topography. More conversion to these baroclinic modes decreases the energy left over for propagation of the barotropic tide, resulting in lower amplitudes at the coast. By contrast, in very shallow regions (e.g., North Sea[25], Yellow Sea[26]), near-surface warming can stabilize rotating barotropic flows against turbulent dissipation[21] and thus enhance the tide's elevation signal. A key issue when representing these processes in tidal simulations is that they pose tight requirements on time step and domain discretization.

In this work, we use a global internal-tide permitting numerical model to quantify how recent changes in ocean stratification have affected tidal surface elevations, primarily $M_2$ and primarily its barotropic component. Emphasis is on trends from 1993 to 2020, a period where we have suitable three-dimensional (3D) density data. Our simulations suggest that strengthening of stratification during that period (Fig. 1) induced coastal $M_2$

[1]Institute of Geodesy and Geoinformation, University of Bonn, Bonn, Germany. [2]Geodesy & Geophysics Laboratory, NASA Goddard Space Flight Center, Greenbelt, MD, USA. ✉e-mail: opel@igg.uni-bonn.de

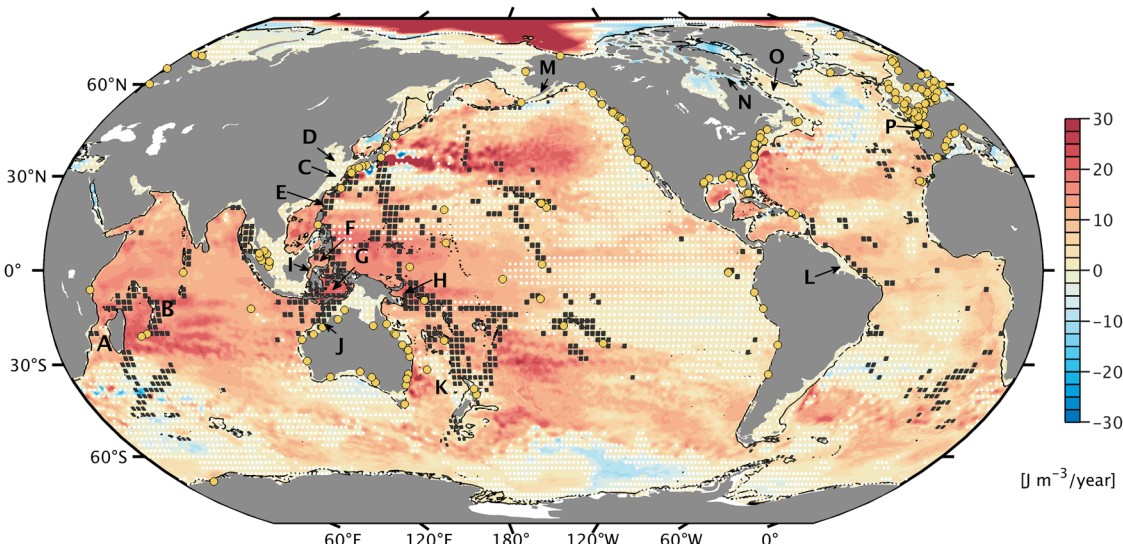

**Fig. 1 | Trends in ocean stratification, 1993–2020.** Color shading illustrates the linear change in potential energy anomaly $\phi^{91}$ (J m$^{-3}$ yr$^{-1}$), calculated from Global Ocean Physics Reanalysis GLORYS12, Version 1[28] annual temperature and salinity profiles. Areas with statistically insignificant trends (95% confidence interval) are stippled white. Black squares highlight 2° × 2° cells where the area-averaged modeled $M_2$ tidal energy conversion rate, $C$, exceeds 3 GW m$^{-2}$ (Methods). Yellow markers indicate the locations of tide gauges used in this study. Geographical names mentioned in the main text are (**A**) Mozambique Channel, (**B**) Mascarene Ridge, (**C**) East China Sea, (**D**) Yellow Sea, (**E**) Luzon Strait, (**F**) Celebes Sea, (**G**) Banda Sea, (**H**) Solomon Sea, (**I**) Makassar Strait, (**J**) King Sound, (**K**) Tasman Sea, (**L**) Amazon Shelf, (**M**) Bristol Bay, (**N**) Hudson Strait, (**O**) Labrador Sea, and (**P**) Bay of Biscay.

amplitude trends of order ~0.1 mm yr$^{-1}$, matching estimates from regional tide gauge networks in approximately half of the 10 considered cases. The coastal $M_2$ trends are part of a concerted global tidal response to increasing stratification, also involving basin-wide changes with a clear tendency for an amplitude decline. Considering additional constraints from satellite altimetry, we argue that the contemporary weakening of $M_2$ in the open ocean and in several marginal seas is caused by increased tidal conversion at shelf breaks and mid-ocean topography. The study adds new facets to the discussion of secular changes in tides and highlights processes that are yet to be included in physics-based projections of extreme sea levels.

## Results and Discussion
### Basic assessment
We use a $\frac{1}{12}$° setup of the Massachusetts Institute of Technology general circulation model[27] (MITgcm) to propagate the $M_2$, $S_2$, $K_1$, and $O_1$ constituents through annually varying density structures from 1993 to 2020 (Methods). Each of these 28 simulations is strongly relaxed to the respective year's mean stratification, taken from an eddy-resolving ocean reanalysis[28]. This approach to modeling eliminates the need for a costly multi-decadal simulation with continuous buoyancy and momentum forcing by the atmosphere. We extract harmonic constants in surface elevation (i.e., amplitude and Greenwich phase lags) of the four constituents from each forward integration and perform a validation against selected in situ data and altimetry-constrained tidal models for one run (Supplementary Table 1). These comparisons indicate high accuracy of the tidal solutions in the deep ocean (e.g., $M_2$ mean square error = 4.9 cm) and reduced, but still sufficient fidelity in areas shallower than 1,000 m, similar to other non-assimilative global baroclinic tide models[29,30].

Our simulations develop a realistic internal tide field, featuring familiar regions of generation, e.g., near Madagascar, Luzon Strait, Hawaii, French Polynesia, the Amazon Shelf, or around the Indonesian Seas; see Supplementary Fig. 1 and black squares on Fig. 1. Over the 40 days of integration, low-mode internal tides travel O(1,000 km) in characteristic beams, which gradually decay as waves break in the interior and at the boundaries of the ocean. A simple evaluation in terms of area-averaged stationary internal tide amplitudes[31] in a number of hot spot regions (Supplementary Table 1) shows that the modeled wave field compares well with an altimetry-based solution[32], although diurnal baroclinic tidal amplitudes are overestimated near generation sites. The globally integrated barotropic-to-baroclinic energy conversion rate ($\overline{C}$) over all constituents is 0.70 TW, a considerable fraction of the total tidal energy dissipation, $D$ = 3.42 TW. These estimates are dominated by contributions from $M_2$ ($\overline{C}$ = 0.50 TW, $D$ = 2.41 TW).

### Sensitivity to stratification changes
Figure 2 shows the total root mean square (RMS) variability of the $M_2$ surface tide computed from the yearly amplitude and phase maps (Methods). We neither separate temporal components (i.e., trends vs. interannual changes) nor spatial scales (i.e., barotropic vs. baroclinic tides) at this stage. The picture is evidently dominated by temporal variability of the baroclinic tide, reflecting the impact of stratification changes on wave propagation characteristics, modal partitioning, and variations in source strength, as documented for numerous places[22,33–38]. Focusing on spatial structures indicative of internal tides, we find peak RMS signals of up to 3 cm in the Celebes Sea, ~2 cm in the Banda and Solomon seas[39], and ~1–1.5 cm near generation sites in the Indian Ocean. The simulations also produce structured internal tide variability throughout the Northwest Atlantic. However, this feature might well be a peculiarity of our modeling (particularly the relaxation) approach, as internal tide beams in that area are understood to interact with the Gulf Stream and its mesoscale meanders[40,41]. Similar arguments may hold elsewhere, e.g., for the branching of baroclinic fluxes off the Amazon Shelf[42].

In most hot spot regions, the total $M_2$ RMS variability in Fig. 2 amounts to ≲10% of the mean internal tide amplitude in proximity to generation sites, but increases to magnitudes comparable to the mean amplitude ≳1,000 km away from the sources. The dominant part of this remote non-stationarity rests with the internal tide phase modulation by time-variable background flows[43], as seen in previous modeling work[34,44]. Our map for the non-stationarity of $S_2$ (Supplementary Fig. 2a) bears close resemblance to that of $M_2$, although RMS values are smaller by a factor ~2. For $K_1$ and $O_1$, interannual variability in the deep ocean is largest (5–10 mm) in association with internal tides in the Celebes, Sulu, and Philippine seas; see Supplementary Figs. 2b and 2c. These results usefully complement empirical mapping results for temporal variability of the baroclinic tidal sea level[45,46]—a critical factor when interpreting new-generation wide-swath altimetry data in terms of balanced ocean submesoscale flows[47].

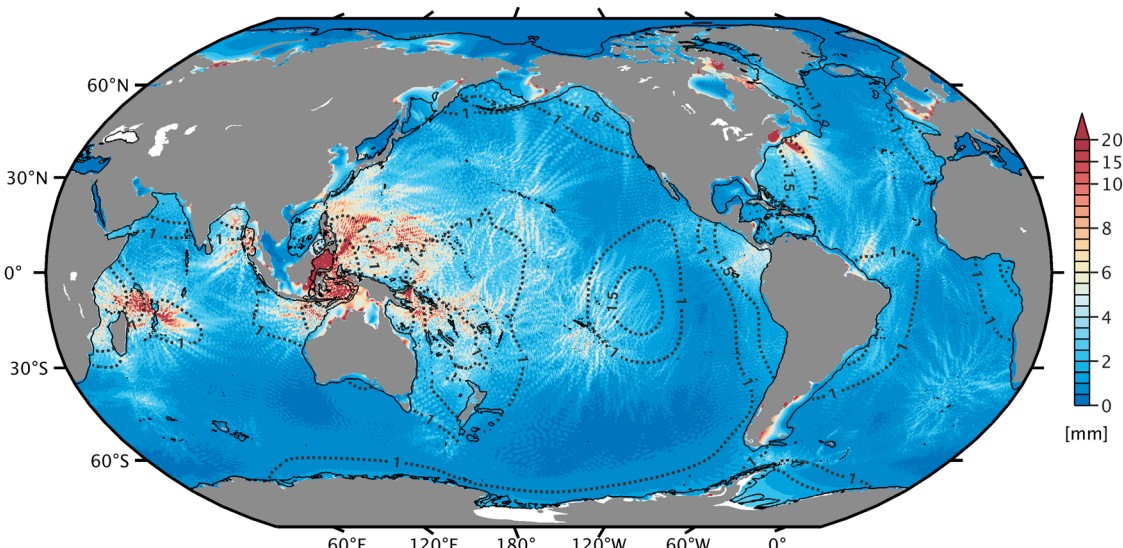

**Fig. 2 | Modeled variability of the $M_2$ surface tide, 1993–2020.** Shown is the total RMS variability[44] (mm), resulting from changes in tidal amplitude and amplitude-weighted phase of the stationary $M_2$ surface tide across the 28 yearly simulations with varying stratification. Linear trends are not removed and therefore contribute to the figure. The black solid line marks the 500 m bathymetry level, and dotted contours (1 and 1.5 mm) indicate the total RMS variability of the barotropic tide alone (drawn only for depths >500 m).

Figure 2 also reveals a sensitivity to stratification changes in the barotropic $M_2$ tide (now separated from the baroclinic tide, see Methods). The effect is most clearly seen in shallow, tidally active regions, where the year-to-year $M_2$ variations amount to a total RMS of 1–3 cm, approximately 1% of the tidal amplitude. Prominent examples include the English Channel, Hudson Strait, Bristol Bay north of the Aleutian Islands, the West Florida and Patagonian shelf areas, Makassar Strait east of Borneo, the coastline of China (to some degree), and numerous embayments on the northern Australian seaboard. Values of order 5 cm are confined to the Bay of Fundy, Canada, and King Sound in Northwest Australia. Most of these locations are known to host a pronounced seasonal $M_2$ cycle, which has been linked to diminished levels of vertical mixing in stably stratified summer waters[25]. Accordingly, changes in turbulent dissipation might account for part of the simulated low-frequency variability in shallow regions—though there are cases where conversion effects are known to dominate (e.g., the Gulf of Maine[22]). Away from shelf seas and coastal waters, the total RMS variability in Fig. 2 shows long-wavelength features of order 1–2 mm (dashed contours), again manifesting a sensitivity to stratification changes in the barotropic tide. These background signals emerge from the fog of baroclinic tidal variability in a few regions, particularly in the eastern South Pacific Ocean. Below we map trends of the simulated barotropic surface tide in the global ocean more clearly, based on appropriate filtering of the baroclinic tidal component (Methods).

## $M_2$ trends—global synthesis

Figure 3 brings together several important elements of our study. We plot trends of the barotropic $M_2$ amplitude, deduced from the 3D MITgcm simulations with annually varying density structures (Fig. 3b), along with another trend estimate from a 2D shallow-water model[19], accounting for the tidal response to relative sea level rise (Fig. 3c). Note that the time spans underlying these maps are identical (1993–2020) and that the linear rates extracted from the MITgcm are corrected for residual trends caused by varying amounts of steric expansion across the simulations (Methods, Supplementary Fig. 3). This global depiction of the tidal response to two distinct physical processes is complemented in Fig. 3a with long-term trends of the $M_2$ amplitude as deduced from TOPEX/Poseidon (T/P) and Jason satellite altimetry data. We flexibly bin the along-track data in space (in part to avoid contamination by internal tides) and solve for linear rates of the $M_2$ in-phase and quadrature components, again over 1993–2020 and within

the $\pm 66°$ T/P latitude range. For legibility, we smooth the original trend estimates (Supplementary Fig. 4) using a boxcar filter with varying size ($10° \times 10°$ in deep water).

The altimetry solution in Fig. 3a is not corrected for all possible systematic errors (see "Methods") but internally consistent enough to reveal large-scale trends in the $M_2$ amplitude throughout the major ocean basins. The trends are primarily negative and attain magnitudes of ~0.1 mm yr$^{-1}$ (more in some locations), as seen in estimates by Bij de Vaate[12] at individual satellite crossovers. Importantly, our baroclinic model simulations (Fig. 3b) reproduce many of the satellite-based $M_2$ changes in terms of structure, sign, and—to some extent—their amplitude. Focusing on trends exceeding the 68% confidence level, the correspondence is particularly evident in the tropical Indian Ocean (including the Mozambique Channel), off the western coast of South America, the greater Gulf of Alaska, the Labrador Sea, and the Northeast Atlantic around the Bay of Biscay. By contrast, trend patterns are largely disparate between Fig. 3a and b in the eastern Pacific and the (sub-)tropical Atlantic. Sea level rise appears to be a relevant forcing of $M_2$ changes in the latter region; cf. Fig. 3c. In the Tasman Sea, strong negative trends in the altimetry (up to −0.3 mm yr$^{-1}$, significant at ≥ 99% confidence) exceed estimates from the MITgcm by a factor of ~4, but independent data (e.g., from a network of tide gauges) would be needed to shed better light on the matter.

We complement this somewhat qualitative assessment with Table 1, a comparison of area-averaged $M_2$ amplitude trends from altimetry and the baroclinic simulations in four regions with widespread negative and largely robust trends (Tropical Indian Ocean, Tasman Sea and Pacific waters east of New Zealand, Northeast Pacific, Northeast Atlantic). Our simulations typically capture 60% of the satellite estimate, pointing to a residual that may arise from model limitations, subtle errors in the altimetry solution, or both (Methods). As might be expected from Fig. 2, interannual variability modulates the regional $M_2$ decline in the runs with stratification changes, see Supplementary Fig. 5. In particular, three of the four-time series of area-averaged annual $M_2$ anomalies feature a pronounced trough over ~1997–2004, possibly in relation to large-scale climate modes (e.g., Pacific Decadal Oscillation).

Taken together, Fig. 3 and associated maps for in-phase and quadrature components (Supplementary Figs. 6–7) show that there are indeed contemporary large-scale trends of $M_2$ in the open ocean and that these signals are primarily caused by changes in ocean stratification rather than by

**Fig. 3 | Barotropic M$_2$ amplitude trends, 1993–2020, from satellite altimetry and numerical ocean models.** Shown are trends (mm yr$^{-1}$) from analysis of **a** T/P-Jason altimetry observations, **b** MITgcm simulations, considering the effect of stratification changes, and **c** barotropic model simulations, considering the effect of relative sea level rise (updated from ref. 19). Changes in the surface manifestation of internal tides have been removed by dedicated processing of the altimetry and baroclinic modeling results (Methods). Note that the color axis in **a** extends to ±0.35 mm yr$^{-1}$, a factor of 1.75 larger than in **b** and **c**. Heavy (or light) black dots identify regions where values do not pass the 68% (or 95%) level for statistical significance. Areas considered for the trend comparison in Table 1 are outlined by black polygons in **b**.

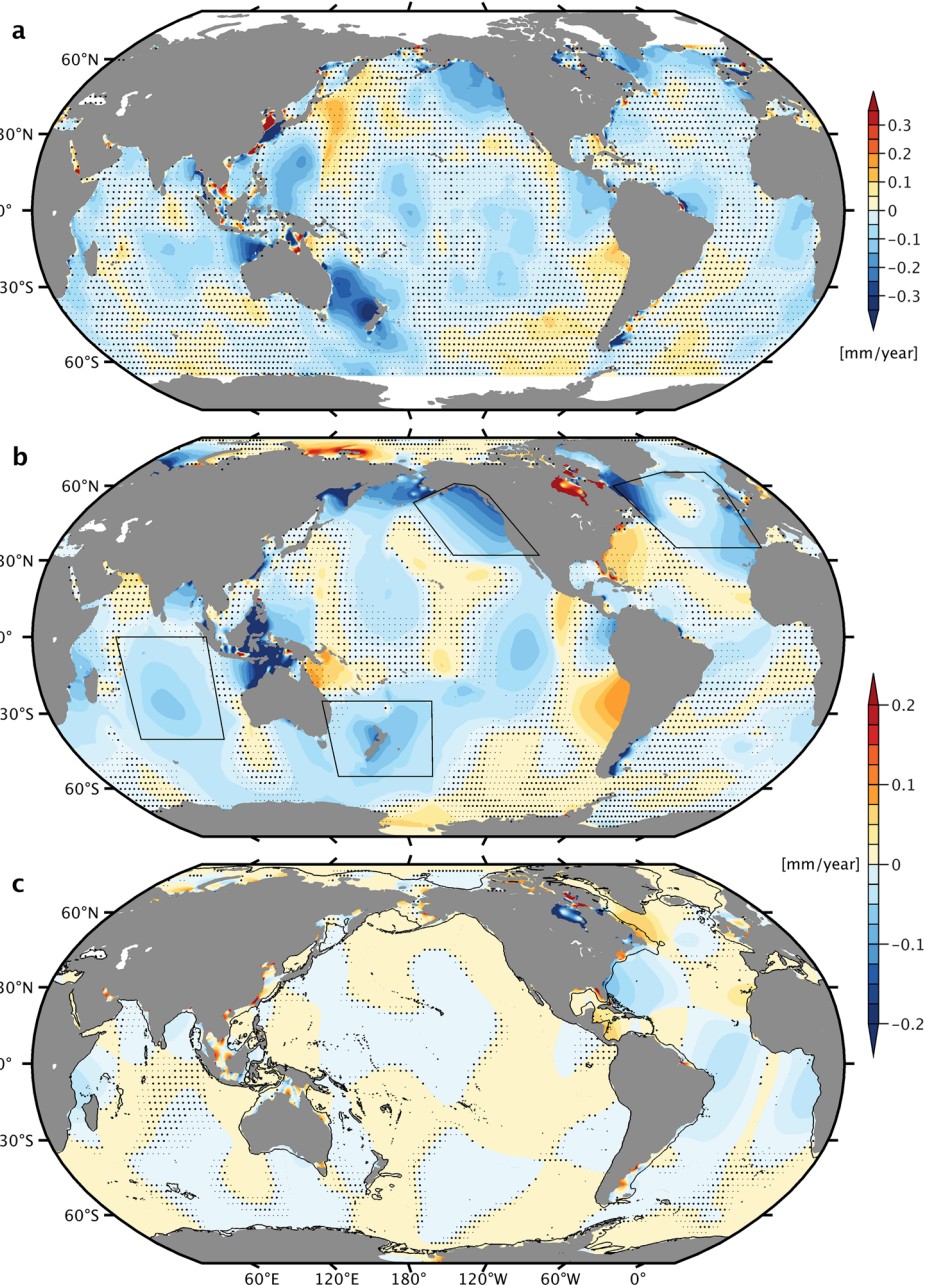

relative sea level rise; cf. the small open-ocean trends in Fig. 3c. The general tendency toward decreasing M$_2$ amplitudes is consistent with the observed stratification increase over past decades[24] (Fig. 1), as sharper vertical density

## Table 1 | Area-averaged M$_2$ amplitude trends (mm yr$^{-1}$) between 1993 and 2020 in four selected regions

|  | Altimetry | 3D model |
|---|---|---|
| Tropical Indian Ocean | − 0.06 ± 0.00 | − 0.04 ± 0.01 |
| Tasman Sea/New Zealand waters | − 0.14 ± 0.00 | − 0.05 ± 0.01 |
| Northeast Pacific | − 0.13 ± 0.01 | − 0.08 ± 0.01 |
| Northeast Atlantic | − 0.08 ± 0.01 | − 0.05 ± 0.01 |

Barotropic M$_2$ amplitude trends, averaged within the black boxes in Fig. 3b, from satellite altimetry and model simulations with time-varying stratification. 68% confidence intervals are provided. All estimates are statistically significant at the 99% level. M$_2$ trends associated with sea level rise over the 1993–2020 period are zero to the second digit except for the Northeast Atlantic and Pacific (0.01 mm yr$^{-1}$, statistically significant at ≥ 99% confidence).

gradients are likely to enhance the conversion of barotropic to baroclinic wave energy in areas of rough topography. Indeed, recent altimetry analysis[38] suggests strengthening of the mode-1 M$_2$ internal tide kinetic energy over 2010–2019 compared to 1995–2009, by ~7% on a global average. This estimate is dominated by contributions from a few regions (the Mascarene Ridge near Madagascar, Luzon Strait, the West Pacific and Aleutian trenches), all featuring increases in the mode-1 surface amplitudes of ~2–3 mm over 12.5 yr. Trends fitted to the internal tide in our simulations (Fig. 4) suggest similarly distributed strengthening rates (~0.3 mm yr$^{-1}$, significant at ≥ 95% confidence), also near French Polynesia and the Amazon Shelf. To the extent the enhanced baroclinicity manifests a source effect, and not just more efficient wave propagation, past decades have seen an increase in tidal conversion and thus the dissipation of barotropic wave energy in the deep ocean. More dissipation (i.e., effective dampening) of resonantly excited normal modes near the M$_2$ frequency[48] provides a feasible explanation as to why barotropic amplitude trends in Fig. 3 are mostly negative and structured in space along normal mode features.

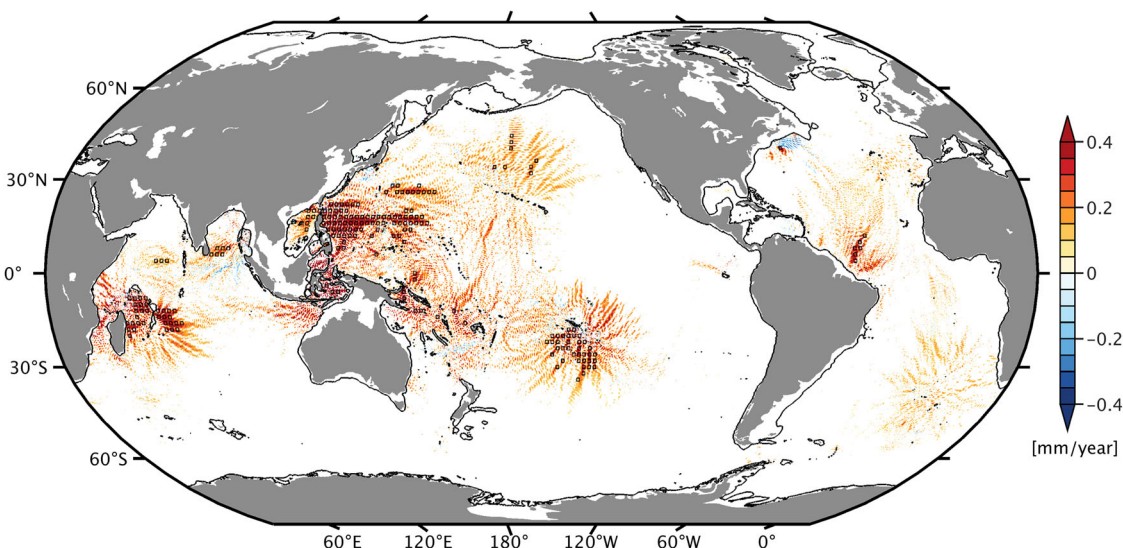

**Fig. 4 | Modeled trends of the M$_2$ internal tide amplitude, 1993–2020.** Trend values (mm yr$^{-1}$) are for the stationary M$_2$ internal tide in sea surface height on a $^1/_{12}$°grid, as deduced from the MITgcm simulations by subtracting the derived barotropic M$_2$ trends in in-phase and quadrature components (Supplementary Figs. 6–7) from the linear rates fitted to the 28 yearly surface tide solutions. Grid points with statistically insignificant trends (at the 68% confidence level) and areas shallower than 500 m are masked. Black boxes are drawn for 2° × 2° cells where trend values are significant at 95% confidence for at least a third of the contained $^1/_{12}$°grid points.

As we move from the open ocean into shallow water, interpretation of Fig. 3 becomes somewhat involved. M$_2$ attains shorter length scales and responds appreciably to sea level rise[17], as apparent, e.g., in the Indonesian seas. At the same time, confidence in the altimetric solution decreases, in part due to the potential for aliased tidal variability in the prior for non-tidal ocean signals[16] (Methods). Despite the caveats, it is hard to dismiss the large observed M$_2$ trends in the East China and Yellow seas (± 1.0–1.5 mm yr$^{-1}$, significant at ≥ 99% confidence) as artefact. In particular, the structure and magnitude of the signals in the altimetry map are consistent with regional tide modeling results in ref. 10 (Figure 9) and ref. 49 (Figure 6F). These works suggest that large-scale tidal flat reclamation along the Jiangsu Coast (north of Shanghai) since the middle of the 20th century induced major changes in wave propagation characteristics, accompanied by a 5–10 cm amplitude increase (decrease) in the western Yellow Sea (East China Sea). Whether and by how much these coastline developments altered tidal dissipation in the area is at present unknown. The question certainly merits consideration, given that changes of tidal dissipation in shelf seas can exert a back-effect upon open-ocean tides[50], albeit weak in the case of a non-resonant and strongly damped shelf. Global barotropic tide simulations with suitable perturbations of coastline positions could help quantify such back-effect and its possible signature in the altimetry trend estimates over the West Pacific Ocean (Fig. 3a).

Elsewhere in a number of marginal seas and continental shelf regions (e.g., Celebes Sea, North Australian Basin, southern Patagonian Shelf, Northwest European Shelf, and parts of the Amazon Shelf), we again find approximate agreement between M$_2$ amplitude trends in the baroclinic model and those inferred from satellite altimetry. In all of these cases, the trends are negative (from about −0.1 to −0.5 mm yr$^{-1}$), at odds with the supposed amplitude increase if the underlying cause were reduced turbulent energy losses (Introduction). Instead, we can make another case for barotropic-to-baroclinic conversion: stratification strengthened just outward of the continental shelf (Fig. 1), including slope regions that scatter parts of the incident tidal energy into baroclinic motions. Enhanced conversion at some of those places is indeed implied by the observed M$_2$ mode-1 amplification over past decades[38]. The result ought to be concomitant weakening of the barotropic tide.

**Regional foci**

We now shift the focus toward the coast and compare our simulation results against measured M$_2$ amplitude trends at tide gauges. The analysis is restricted to a few regions with sufficient density of suitable ground truth data, viz., Australia and Southeast Asia (Fig. 5a), Europe (Fig. 5b), and North America (Fig. 6). The selected stations form a subset of the network used for basic model validation (see Fig. 1) and exclude locations where the surface signature of the internal tide is large enough to appreciably impact the observed M$_2$ changes (e.g., at West Pacific islands or the Ryukyu Arc between Japan and Taiwan). We account for non-equilibrium values of the 18.61-year nodal modulation and estimate amplitude trends for the exact same period as in our numerical modeling (1993–2020). The simulated trends in Figs. 5 and 6 represent the sum of the M$_2$ response to stratification changes (Fig. 3b) and sea level rise (Fig. 3c). We refrain from adding the altimetry solution to the comparison but note that there are indeed regions where it is in accord with the in situ estimates (e.g., along the Mid-Atlantic Bight or the British Columbia Coast, Canada).

Both tide gauge and model results point to distinct regional structures in coastal M$_2$ trends since 1993. The spatial extent and sign of these changes generally differ from those reported in previous tidal surveys[1,3,19], emphasizing that any such analysis is sensitive to the adopted time span, as well as to changes in instrumentation. The model-data agreement varies from being very poor or merely qualitative (for example, in the Skgaerrak/Kattegat region, and around the Great Barrier Reef) to a much tighter correspondence in Northwest Australia or along the Northeast Pacific coastline. Occasional inconsistencies between neighboring tide gauges, possibly due to local effects, are also part of the picture. To better encapsulate these findings, we form a budget of contemporary M$_2$ amplitude trends in 10 selected coastal regions (Fig. 7), obtained by averaging measured trends at available tide gauges in predefined patches (Supplementary Figs. 8–9) and comparing them with the averaged simulated trends, sampled at the same locations. We omit regions where neither models nor observations indicate a robust, spatially coherent M$_2$ trend (e.g., in Japan and the western Gulf of Mexico). A breakdown of modeled signals by processes, represented as pie charts, complements Fig. 7.

This synthesis reiterates the role of stratification in driving long-term M$_2$ changes in the Gulf of Alaska, along the US/Canadian West Coast, and on the Northwest Australian continental shelf. For the latter two regions, budgets are balanced within a factor of 1.5. The disparity is notably larger in the Gulf of Alaska (modeled −0.13 ± 0.01 mm yr$^{-1}$ vs. observed −0.42 ± 0.03 mm yr$^{-1}$, 68% confidence intervals), in part due to an excessive negative trend at one particular station (−1.1 mm yr$^{-1}$ at Queen Charlotte

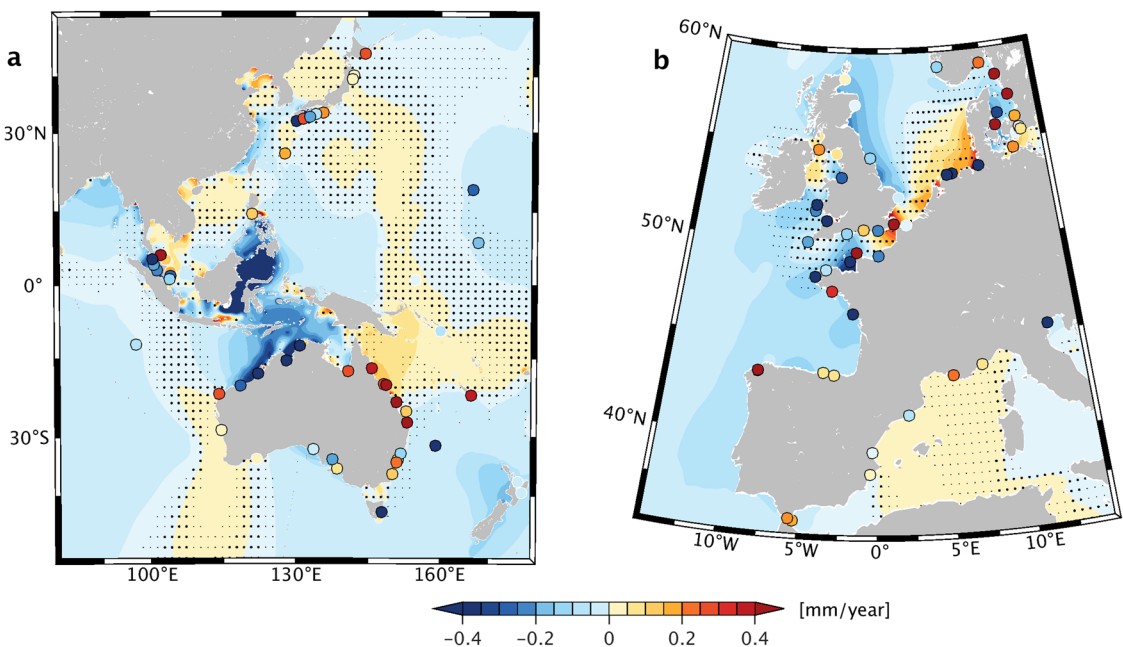

**Fig. 5 | M$_2$ amplitude trends around Australia/Southeast Asia and Europe, 1993–2020.** Colored markers represent measured M$_2$ trends (mm yr$^{-1}$) at tide gauge locations in **a** Australia and Southeast Asia, and **b** Europe. Markers are highlighted with black (or respectively white) outlines wherever fitted rates are statistically significant (insignificant) at 68% confidence. Color shading indicates modeled amplitude trends, representing the combined response of the barotropic M$_2$ tide to stratification changes (Fig. 3b) and relative sea level rise (Fig. 3c). Heavy (or light) black dots indicate areas where model trends do not pass the 68% (or 95%) threshold for statistical significance.

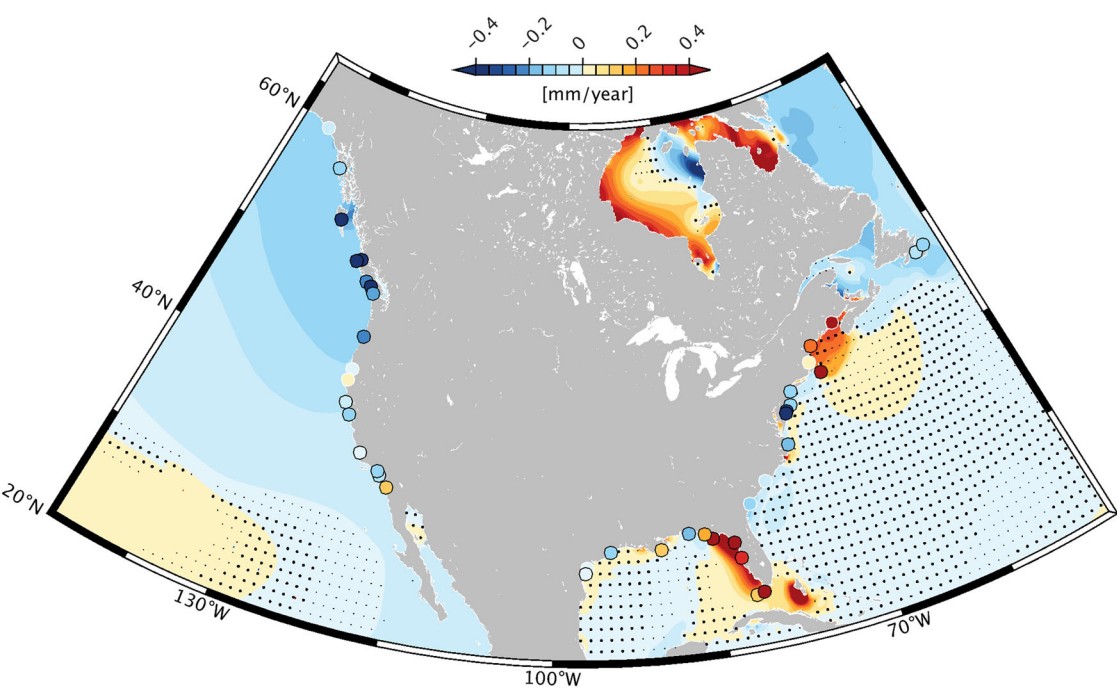

**Fig. 6 | M$_2$ amplitude trends around North America, 1993–2020.** As in Fig. 5 but for the ocean and marginal seas encasing North America. Heavy (or light) black dots indicate areas where model trends do not pass the 68% (or 95%) threshold for statistical significance. Colored markers represent measured M$_2$ trends (mm yr$^{-1}$) at tide gauge locations, highlighted with black (or respectively white) outlines wherever fitted rates are statistically significant (insignificant) at 68% confidence.

City on Graham Island; cf. Supplementary Table 2). The value is inconsistent with the linear M$_2$ change at the next higher-quality tide gauge further north (Sitka, Alaska, see Supplementary Fig. 10) but is left in the average to provide some indication of the uncertainty involved in our analysis. The MITgcm simulations also suggest a negative amplitude trend ($\sim -0.36 \pm 0.07$ mm yr$^{-1}$) in the Strait of Malacca, west of Malaysia, likely

linked to the M$_2$ decrease at the entrance of the strait, and beyond. Observations reveal a near identical decrease ($\sim -0.39 \pm 0.13$ mm yr$^{-1}$), albeit drawn from three relatively short tide gauge series. In Northeast Australia, the combined modeled tidal response to sea level and stratification changes captures the sign of the observed amplitude increase, yet only 1/5 of its magnitude. This mismatch may indicate model representation errors

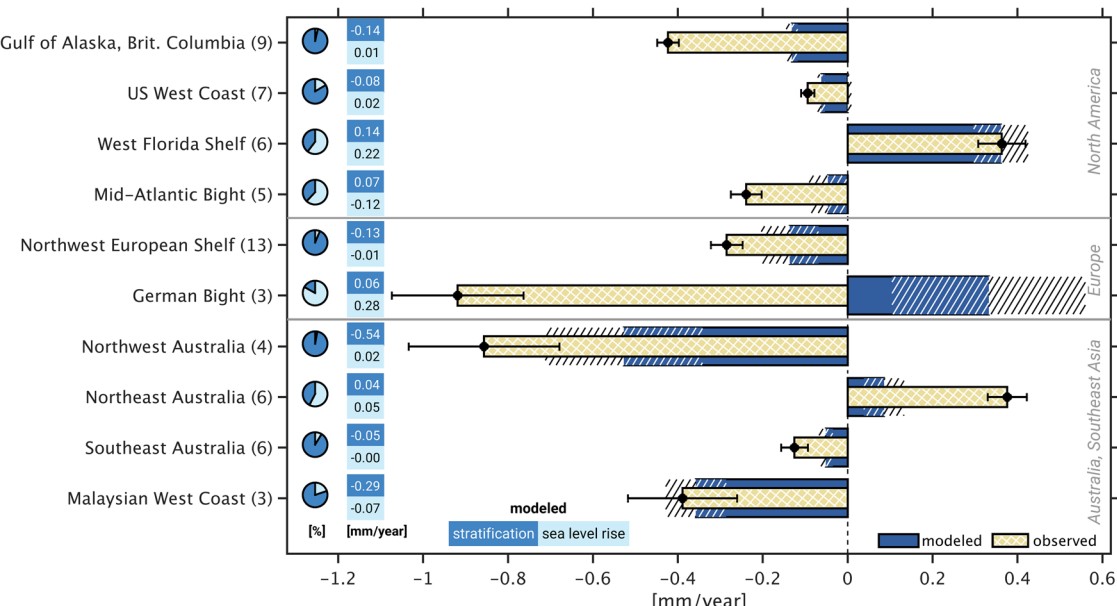

**Fig. 7 | Budget of contemporary M$_2$ amplitude trends in selected coastal regions.** Shown are spatial averages of M$_2$ amplitude trends (mm yr$^{-1}$) at tide gauge locations in 10 regions, deduced from water level observations (cross-hatched creme bars) and numerical modeling results (Fig. 3) that account for the combined effect of stratification and sea level changes over 1993–2020 (dark blue bars). Black error bars and the hatched extension of the modeled M$_2$ trends represent the respective 68% confidence limits. Only tide gauges with observed trends being likely significant (68% level) are considered. Numbers in parentheses on the vertical axis show the total count of tide gauges per average. Pie charts on the left indicate the relative contributions of the two different driving processes to the modeled M$_2$ amplitude trend in each region. The region referred to as "Northwest European Shelf" comprises the Celtic and Irish seas, and the English Channel (see Supplementary Fig. 8b).

(related to, e.g., geometry and seabed roughness changes[5]) in a very shallow, tidally active environment.

Large formal errors on model trends tend to impede broader conclusions on the causes of contemporary M$_2$ trends on both sides of the North Atlantic, including adjacent marginal seas. To the west and south of the United Kingdom ("Northwest European Shelf" region, Fig. 5b), we find hints for another amplitude decrease (modeled $-0.14 \pm 0.07$ mm yr$^{-1}$ vs. observed $-0.29 \pm 0.04$ mm yr$^{-1}$). However, the MITgcm results in that region are no more than tentative, as they are overly sensitive to the correction for steric expansion effects (see "Methods"). To the other side of the basin, marked year-to-year M$_2$ variations in the runs with stratification changes (Fig. 2) overprint any trend in the Gulf of Maine (not included in Fig. 7). We nevertheless note that the red color (i.e., strengthening of M$_2$ by $\sim$0.20 mm yr$^{-1}$) inside the Gulf in Fig. 6 points to weakening of stratification over the nearby continent slope[22]. The argument is further substantiated by the negative trends in the modeled internal tide off the Scotian Shelf (Fig. 4). Elsewhere along the US Atlantic and Gulf coast, simulated M$_2$ changes are the result of competing sea level and stratification effects, compare also Fig. 3b and c. In the Mid-Atlantic Bight, the amplitude trends from the two forcings are subject to cancellation, rendering the simulated net signal indistinguishable from zero (and five times smaller than the tide gauge estimate). By contrast, both processes add up to induce a comparatively large M$_2$ increase ($0.36 \pm 0.06$ mm yr$^{-1}$) on the West Florida Shelf, matching observations within formal uncertainties. Whether the positive contribution from stratification reflects a decrease in tidal conversion on the nearby Florida Escarpment, or rather the result of reduced turbulent dissipation[21] due to surface warming in coastal waters, remains to be explored.

Our work provides little insight as to the exact processes occurring in the German Bight. The M$_2$ amplitude increased steadily throughout the second half of the 20th century[7,19], but analyses of more recent water level data[12] (see also Supplementary Fig. 10) suggest that it is now declining (since 2014 to be precise). Over 1993–2020, the observed trend, taken as the average of the three tide gauges in Fig. 5b, amounts to $-0.92 \pm 0.16$ mm yr$^{-1}$, commensurate with altimetric estimates in the area[12]. Sea level rise cannot account for the decline, as greater water depths are supposed to increase the

amplitude of a Kelvin wave at the far end of a semi-enclosed rotating basin[51]. Instead, the negative trends in the southern North Sea point to increased tidal dissipation near the coast[19,52], caused, e.g., by silting and sediment accumulation at rates that exceed mean sea level rise, especially in the Wadden Sea[53]. In any case, the recent trend reversal of M$_2$ around the German Bight contextualizes the notion of long-term changes in ocean tides and suggests that several, possibly competing processes might be at work. The region therefore remains an obvious target for more dedicated numerical modeling efforts.

## Conclusions

Joint use of observations and model results has allowed us to sketch a tentative yet first coherent picture of trends in the global ocean tides since 1993. As is clear from Fig. 3 and Figs. 5–7, stratification changes are a crucial element of that picture, across basins and from deep-ocean to coastal realms. The main mechanism by which the recent strengthening of stratification (Fig. 1) affects the barotropic tide appears to be enhanced tidal conversion over steep topography. In some cases, e.g., northwest of Australia, greater fluxes into internal tides occur at the shelf break (see Fig. 4), emphasizing why tide gauges shoreward in the direction of the wave propagation are indicating a persistent decrease in tidal amplitude. Our focus in these considerations has almost exclusively been on M$_2$. There are nevertheless some interesting features in the trend maps of other constituents (Supplementary Fig. 11), including a $\sim$$-0.20$ mm yr$^{-1}$ decrease of the S$_2$ amplitude in the Indonesian Seas, and a similar decrease of O$_1$ ($-0.10$ to $-0.20$ mm yr$^{-1}$) throughout the South China Sea and the Gulf of Thailand. The latter is also evident in a preliminary altimetric O$_1$ trend estimate by us and may well point to intensified barotropic-to-baroclinic conversion near Luzon Strait, a major chokepoint for O$_1$ energetics[54].

Our results raise questions over the treatment of ocean tides as exact stationary phenomena in space-geodetic analyses and products derived thereof (e.g., models of sub-daily Earth rotation variations[55] or coastal altimetry products). In particular, any large-scale temporal variability of the barotropic tide adds to the uncertainty of background models employed for de-aliasing satellite gravimetry observations. In this regard, Fig. 2 is a useful

guide for incorporating such variability into the stochastic component of the gravity field determination[56]. Whether or not more explicit handling of tidal changes in satellite gravimetry or altimetry processing is warranted remains to be assessed, but one can expect the matter to be reinforced by the ongoing and projected acceleration in upper-ocean warming[57]. The most attractive target emerging from our work, however, is to revisit global projections of future extreme sea levels[13,58], which presently neglect changes in tides or consider them as a function of sea level rise alone.

## Methods

### Modeling approach

We model tides with a global MITgcm[27] configuration that solves the three-dimensional primitive equations, under the hydrostatic and Boussinesq approximations. The dynamical equations are discretized on a $\frac{1}{12}°$ curvilinear grid referred to as LLC1080 (latitude-longitude-polar-cap grid, with 1080 points along one-quarter of the Earth's circumference at the equator). As with any other realization of the LLC family[59], LLC1080 consists of a latitude-longitude sector between 70°S and 57°N, with grid refinement in the tropics. At high latitudes, transitioning to a 2D conforming mapping algorithm for spherical geometry is allowed for, resulting in an Arctic cap in the northern hemisphere. Note that a nominal grid spacing of $\frac{1}{12}°$ is standard in global modeling of tidal changes[19] and also sufficient to resolve the generation and propagation of low-mode internal tides in the deep ocean[60]. In the vertical, our setup consists of 59 $z$ levels with spacings that vary from 6 m near the surface to 484 m at the deepest level (7130 m). Bottom topography is based on the 30-arcsec RTopo-2 dataset[61] and represented in the model by a partial step formulation[62]. Our configuration uses an implicit linear free surface and a default time step of 75 sec for both momentum and tracer equations.

We do not attempt to model changes in tides by means of a single (and inordinately costly) multi-decadal simulation. Instead, the analysis period 1993–2020 is partitioned into 28 separate simulations of short duration. Each of these time slice simulations is started from rest and integrated for 40 days under identical forcing, yielding the mean tidal solution of a particular year after harmonic analysis of the last 15 days (cf. refs. 22,34 for similar approaches). As we relax the model to the assumed true stratification in each year (see below), atmospheric forcing is omitted. The only forcing applied is the equilibrium tidal forcing of four primary constituents ($M_2$, $S_2$, $K_1$, $O_1$, with solid Earth tide correction) and the corresponding self-attraction and loading (SAL) tide. We compute these external SAL fields ourselves, by applying a spherical harmonic formulation[63] to each partial wave's in-phase and quadrature components from the TPXO9-atlas[64] (updated version). Nodal variations are also omitted from the model forcing, as their inclusion would unnecessarily complicate comparisons of the yearly simulations amongst each other and with tide gauge observations.

To work out the response of tides to time-varying stratification, we integrate each time slice from 3D initial potential temperature ($\theta$) and salinity ($S$) fields pertaining to that specific year. The ($\theta$, $S$) data are extracted from the Global Ocean Physics Reanalysis GLORYS12, Version 1[28], a high-resolution and eddy-resolving model-data synthesis. GLORYS12 jointly assimilates altimetric sea level observations, in situ temperature and salinity profiles, and remotely-sensed sea surface temperature and sea ice concentration by means of a reduced-order Kalman filter[28]. We average the monthly ($\theta$, $S$) reanalysis fields into annual means and interpolate them horizontally and vertically to the model grid. Throughout each time slice integration, unwanted changes of the background stratification (e.g., by advection processes in geostrophic currents) are suppressed by nudging the evolving potential temperature and salinity fields to their initial values at each 3D location. We use the MITgcm's `rbcs` package to this end and set the relaxation time scale to 3 days. The package is less sophisticated than other relaxation schemes[65,66] that employ frequency thresholds to avoid adverse effects of the nudging (e.g., spurious transient oscillations) on the baroclinic tides. Such effects are nevertheless expected to be small near generation sites (S. Barbot, personal communication, 2023).

Prompted by our work for the Gulf of Maine[22], vertical eddy viscosities and diffusivities are computed via the $K$-profile parameterization[67]. The scheme's background viscosity, accounting for the mixing effect of unresolved breaking internal waves in the momentum equations, takes a value of $5 \cdot 10^{-5}\,\mathrm{m^2\,s^{-1}}$. For horizontal viscosity and diffusivity, we use a modified Leith scheme[68]. Bottom friction is parameterized by a standard quadratic law with a non-dimensional drag coefficient of 0.003. Note that interaction between sea ice and surface tidal currents in polar regions could cause additional variability in tides that is unrelated to stratification (see ref. 25 for an example on seasonal time scales). We therefore switch off the model's sea ice module in all our simulations.

### Limitations of the 3D model

The approximate agreement between observed and simulated $M_2$ changes in a number of regions lends general credence to our MITgcm setup. However, the tendency of most open-ocean trends in Fig. 3b to be ~40% smaller than in the altimetric solution (Fig. 3a) suggests that the model underestimates the presumed increase in tidal conversion, due to, e.g., lack of topographic detail or errors in the background stratification. Particularly conspicuous in this regard is the North Pacific off the Aleutian Ridge, where altimetry shows marked strengthening of the internal tide[38], while the model does not (Fig. 4). Repeats of our experiments, using hydrography from other ocean state reconstructions[69] and employing higher model resolution—both vertically and horizontally—are clearly worth pursuing. In addition, the LLC1080 grid spacing in the tropics (~9 km) is a factor of ~2 too coarse to fully resolve narrow straits and the complex baroclinic tide field in the Indonesian Seas[70]. Given that errors in the simulated baroclinic tides may also feed back to the barotropic tide[70], the strong negative Indonesian $M_2$ trends in Fig. 5a, amounting to $-0.6\,\mathrm{mm\,yr^{-1}}$ in the Celebes Sea and Makassar Strait, should be treated with caution. Lastly, the adopted $z$ level spacing of 6 m near the surface is relatively coarse, which might affect the model's ability to capture changes in vertical eddy viscosity in very shallow regions (e.g., the German Bight[25])

### Internal tides and related quantities

Baroclinic tidal signals are germane to a number of aspects in this work, including the diagnosis of tidal energy conversion rates (Fig. 1) and the extraction of the barotropic component from harmonically analyzed surface heights (Results section). Dedicated calculations are performed for one simulation (2006, somewhat arbitrarily chosen), where we enabled hourly 3D output of instantaneous zonal and meridional velocities $\mathbf{u} = (u, v)$, along with ($\theta$, $S$) diagnostics to compute the density $\rho$. Provided harmonic analysis of $\mathbf{u}$ at each vertical level, the barotropic velocity $\mathbf{U} = (U, V)$ associated with a particular constituent at a location is

$$\mathbf{U}(z, t) = \frac{1}{H} \int_{-H}^{0} \mathbf{u}(z, t)\mathrm{d}z \qquad (1)$$

where $z$ represents height, $H$ is the resting water depth and $t$ denotes time. Defining the wave-induced density perturbation with respect to the tidal period mean $\langle \cdot \rangle$ as $\rho'(z, t) = \rho(z, t) - \langle \rho \rangle(z)$, the baroclinic pressure anomaly follows from[71,72]

$$p'(z, t) = -\frac{1}{H} \int_{-H}^{0} \int_{z}^{0} g\rho'(\hat{z}, t)\mathrm{d}\hat{z}\mathrm{d}z + \int_{z}^{0} g\rho'(\hat{z}, t)\mathrm{d}\hat{z} \qquad (2)$$

where $g$ is the gravitational acceleration. One can combine $\mathbf{U}$ and the perturbation pressure at the bottom $p_b'(t) = p'(z = -H, t)$ to estimate the depth-integrated barotropic-to-baroclinic energy conversion rate[73]

$$C \approx -\langle \nabla H \cdot \mathbf{U}(t)p_b'(t) \rangle \quad [\mathrm{W\,m^{-2}}] \qquad (3)$$

at bathymetric gradients ($\nabla H$) reckoned in eastward and northward direction. Values for $\overline{C}$ given in the main text represent global integrals of the local energy conversion rate $C$.

Knowledge of the baroclinic tidal bottom pressure anomaly also allows one to compute the surface expression of the modeled internal tide ($\eta'$) as

$$\eta'(t) = \eta(t) - \frac{p_b(t) - p'_b(t)}{\rho_0 g} \qquad (4)$$

using harmonic fits to time series of surface elevation $\eta$ and bottom pressure $p_b$. Here, $\rho_0$ is the constant reference density in the MITgcm. We correct the $p_b$ harmonic for contributions from atmospheric pressure forcing, that is, SAL and the equilibrium tide. Equation (4) underlies the $M_2$ and $K_1$ internal tide maps presented in Supplementary Fig. 1. In our main analysis, however, we focus on yearly anomalies of the barotropic surface tide, subjected to trend fitting (e.g., Fig. 3b). These anomalies are obtained by (i) subtracting from each year's in-phase and quadrature components of $\eta$ the respective 1993–2020 average and (ii) smoothing the resultant residual fields in space. Given that subtraction of a time-mean surface tide solution also removes a considerable fraction of the simulated baroclinic tide and its long-wavelength features, the remaining internal tide signals in the yearly $\eta$ maps are relatively easy to suppress. For $M_2$ and $S_2$, we have found a Hamming window with a cutoff wavelength of 390 km (100 km) in regions deeper (shallower) than 500 m to be a reasonable spatial low-pass filter. For diurnal constituents, the cutoff wavelengths are comparably longer (440 and 210 km).

## Global dissipation rates

Estimates of the dissipation rate, $D$ (see "Basic assessment"), are inferred from the global integral for the rate of working of tidal forces on the ocean tide[74,75]. For semidiurnal constituents (frequency $\omega$) we have

$$D = (24\pi/5)^{1/2} G M \widetilde{H} \rho_0 (1 + k'_2) \omega D^+_{22} \sin \psi^+_{22} \quad [\text{W}] \qquad (5)$$

where $G$ is the gravitational constant, $M$ is the mass of the Earth, $\widetilde{H}$ represents the constituent's potential amplitude in length units, $\rho_0$ is a mean seawater density (1035 kg m$^{-3}$), $k'_2$ denotes the degree-2 load Love number, and $(D^+_{22}, \psi^+_{22})$ are the amplitudes and phase lags of the degree-2, order-2 prograde components of the ocean tide. The expression for diurnal constituents is identical to Eq. (5) but requires degree-2, order-1 spherical harmonics $(D^+_{21}, \psi^+_{21})$ and the factor $(6\pi/5)^{1/2}$ instead of $(24\pi/5)^{1/2}$.

## Effects of sea level change

To quantify the $M_2$ response to sea level rise (Figs. 3–7), we revert to an accurate and computationally efficient barotropic tide model[19]. The model solves the shallow water equations on a $1/12°$ latitude-longitude grid of near-global extent (86°S to 84°N), with partial wave forcing for $M_2$ and a time step-wise spectral treatment of SAL effects. Coastlines and bathymetry are based on cell averages of RTopo-2, as in the MITgcm setup described above. The main difference between the two tide models is that the barotropic single-layer approach does not admit internal tides, thus necessitating a parameterization for the barotropic-to-baroclinic energy transfer[19]. We again follow a time slice approach and conduct 28 separate (17-day long) $M_2$ simulations, each with a slightly modified bathymetry to represent the respective year's relative sea level change. The control run, employing unperturbed RTopo-2 water depths, is taken to be 1993. For all other years, we express the change in relative sea level at a specific location as $\Delta h - \Delta c$, where $\Delta h$ denotes the annual mean sea level anomaly (relative to 1993), and $\Delta c$ is the radial displacement of the crust due to glacial isostatic adjustment (GIA, here also relative to 1993). The underlying GIA rates are taken from ICE-6G_C[76], and for annual averages of $\Delta h$ we use $1/4°$ gridded sea level anomalies from multi-mission satellite altimetry data (Copernicus Climate Change Service, C3S, Climate Data Store, 1993 to 2020[77]). Figure 3c in the main text presents linear rates fitted to these 28 barotropic $M_2$ solutions.

A matter of intricacy is that parts of the $M_2$ trends in the MITgcm simulations may be attributed to local trends in the mean surface height (or equivalently water depth), $\eta_0$, rather than to changes in stratification. Differences in $\eta_0$ from run to run generally reflect varying amounts of steric

expansion as the water columns adjust their vertical extent to conform with the prescribed hydrography. From an examination of the resulting mean dynamic topography across the 28 simulations, we find small but non-negligible trends in model sea level, $\dot{\eta}_0$. In regions shallower than 200 m (i.e., regions where tides become sensitive to water depth changes[17]), values of $\dot{\eta}_0$ range from $-1$ to 1 mm yr$^{-1}$, with notable exceptions, e.g., $-5$ mm yr$^{-1}$ in Hudson Bay, $-3$ mm yr$^{-1}$ over polar continental shelf regions, or 1.4 mm yr$^{-1}$ on the Northwest Australian Shelf (Supplementary Fig. 3a). To quantify the corresponding $M_2$ changes, we multiply the map of $\dot{\eta}_0$ by 27 (to represent 27 years of steric expansion) and impose the so derived field as bathymetry perturbation in a repeat of the MITgcm simulation for 2006. Scaling the difference between the barotropic tide from this simulation and the original 2006 $M_2$ solution by 1/27 yields the wanted amplitude trends, see Supplementary Fig. 3b.

Evidently, steric expansion in the MITgcm induces relatively large $M_2$ trends along the US Atlantic coast ($-0.1$ mm yr$^{-1}$) and on the Amazon ($-0.15$ mm yr$^{-1}$) and North Australian ($\pm 0.1$ mm yr$^{-1}$) continental shelves. Still, larger values of $\sim -0.4$ mm yr$^{-1}$ are seen in the Irish and Celtic seas, thus exceeding any genuine $M_2$ trend in the region due to stratification and sea level effects; cf. Fig. 5b. Given this sensitivity to water depth changes, it may be argued that yearly perturbation runs are more appropriate than a single simulation with adjusted bathymetry. However, $\eta_0$ exhibits little interannual variability (standard deviation of $\lesssim 1.5$ mm in relevant regions) and no obvious correlation with the $M_2$ amplitude changes from the very same simulations. Thus, in our comparisons with tide gauges and altimetry, we consistently use the MITgcm-based $M_2$ trends corrected for residual sea level effects as shown in Supplementary Fig. 3b. The small size of year-to-year changes in $\eta_0$ further implies that the enhanced RMS values in shallow water for all four constituents (Fig. 2, Supplementary Fig. 2) indeed manifest a sensitivity to stratification.

## Satellite altimetry

We build on the intriguing initial results of Bij de Vaate et al.[12], who estimated 30-year trends in ocean tides at the ground-track crossover locations of the T/P and Jason satellites. As the authors acknowledged, tidal changes at any given crossover could be dominated by variability in internal tides (Fig. 2). We have therefore taken a more conventional approach[78] by binning tidal residuals in overlapping regions, small enough to obtain adequate spatial resolution yet large enough to help suppress noise and to average across any potential signals from internal tides (typical deep-ocean wavelengths of order 100 km). We use data from the T/P, Jason-1, Jason-2, and Jason-3 satellites, all restricted to their primary ground-track. Although data from many other altimeter satellites are available, it is critical to maintain consistency of spatial sampling throughout the whole analysis time span (1993–2020, corresponding to the time span used in our modeling). The primary T/P-Jason ground-track has a coarse resolution in shallow and marginal seas, so our results apply mainly to the open ocean. The coverage is also limited to the $\pm 66°$ latitude band.

We use the Radar Altimeter Database System (RADS)[79], accepting all default and standard altimeter corrections, including those for ocean tides. In addition, we use gridded sea-surface heights from multi-mission altimetry[80] as an additional correction to remove non-tidal variability in the ocean. Without this adjustment, our results in Supplementary Fig. 4a would be considerably noisier, especially around western boundary currents. We also remove the signals of stationary internal tides by applying a spatial model based on exact-repeat mission altimetry[32]. Given that the prior barotropic tide model removes most of the large-scale tidal variability, analysis bin sizes can be fairly large: sizes vary by latitude, water depth, and distance to coast, but in low latitudes and in deep water, the bins are sufficiently large ($1.5° \times 6°$ in latitude-longitude) to incorporate at least two ascending and two descending tracks. In each analysis bin, we solve for mean in-phase and quadrature components of the $M_2$ tide (relative to the RADS prior), plus their linear trends. We also solve for three other major tides, although those are not discussed here (in any event, no solar $S_2$ trend would be reliable since it is known that T/P and Jason are currently

inconsistent at that frequency[81]). Because in some locations (mostly in marginal seas) the trend estimation could be impacted by anomalous 18.6-yr nodal modulations[82], we also allow for corrections to two nodal sidelines of $M_2$ (Doodson numbers 255.545 and 255.565; the latter is theoretically zero). To obtain the smoothed trend map in Fig. 3a, we apply boxcar averaging to the in-phase and quadrature $M_2$ trends (Supplementary Figs. 6a and 7a) and then convert to amplitude. Our preferred filter settings ($10° \times 10°$ full width in deep water, $1.5° \times 1.5°$ in shallow water, with transition at 500 m) are largely the result of experimentation.

The potential for systematic errors in the altimeter results is high. We here briefly mention three error sources, each of which merits further investigation and a more extensive discussion elsewhere. (1) The standard altimeter dealiasing (dynamic atmospheric) correction, DAC[83], which accounts for ocean variability at periods shorter than 20 days, includes in its present implementation the radiational component of the $M_2$ tide. This raditional component has an erroneous artificial trend, induced by temporal changes of the lunar semidiurnal air pressure tide in the DAC forcing fields. Given that in some locations, the false trends are a considerable fraction of those seen in Fig. 3a, we have adjusted the altimetry-based tidal trends to account for this error. (2) Our correction for non-tidal sea surface height variability[80], which appears critical for reducing noise in the tidal trend estimates, is known to contain small errors from aliased tidal variability[16]. These leakage effects are thought to be mostly due to short-wavelength internal tides, but frequency-wavenumber spectra do reveal power at longer wavelengths indicative of the barotropic tide. It is unclear at present how these errors affect the trends in Fig. 3a. (3) Tidally coherent errors in the satellite ephemerides are now much reduced over those seen in the early altimeter era[84], but they could reappear in the tiny trend signals we are attempting to recover. Moreover, errors in tidal geocenter models[85] and inconsistencies between geocenter models used in the default RADS orbits, could induce false trends in our final results. Preliminary analyses suggest any such errors are of order 0.05 mm yr$^{-1}$ or less for $M_2$, but other constituents are potentially more problematic.

## Tide gauges
Harmonic coefficients estimated at 203 tide gauge stations, mostly open directly to the sea, form important observational constraints in this study. The network is based on an initial, automated screening of the entire GESLA-3 (Global Extreme Sea Level Analysis Version 3[86–88]) database. Requirements imposed on the hourly records concern longest permissible data gaps (20 days, otherwise the entire calendar year is dropped), minimum temporal coverage (28 years[89]), and a condition of at least 15 calendar years of data. Time series passing these criteria are tidally analyzed per calendar year for amplitudes and phases of 67 constituents, using the UTide software package[90]. We choose to apply ordinary least-squares analysis and a colored spectral approach in the computation of formal uncertainties. After fitting and removing the 18.61-year nodal cycle at each location from the annual $M_2$ constants over the entire record length, we restrict the time series to the 1993–2020 window. The final compilation of 203 sites (cf. Fig. 1) is the result of retaining only higher-quality series in denser tide gauge networks, visual checks (also of phases), and emphasizing regions of enhanced amplitude variability in the model. We also ignore a few stations with known bogus $M_2$ signals (e.g., Churchill, Hudson Bay).

## Potential energy anomaly
Estimates of the potential energy anomaly (Fig. 1) from GLORYS12 annual mean potential density ($\rho^*$) profiles are calculated from[91]

$$\phi = \frac{1}{H} \int_{-H}^{0} (\overline{\rho^*} - \rho^*) gz \, \mathrm{d}z \tag{6}$$

where overbar denotes a depth average

$$\overline{\rho^*} = \frac{1}{H} \int_{-H}^{0} \rho^* \, \mathrm{d}z \tag{7}$$

## Total RMS variability
Following ref. 44, we quantify temporal variability in tidal harmonics at a given location using the RMS deviation ($\delta$) of the complex tidal amplitude relative to a mean:

$$\delta^2 = \frac{1}{n} \sum_{j=1}^{n} |A_j e^{i\varphi_j} - \overline{A_j e^{i\varphi_j}}|^2 \tag{8}$$

where $n$ is the number of available harmonics (28) with amplitude $A$ and Greenwich phase lag $\varphi$, $i \equiv \sqrt{-1}$, and the overbar indicates averaging

$$\overline{A_j e^{i\varphi_j}} = \frac{1}{n} \sum_{j=1}^{n} A_j e^{i\varphi_j} \tag{9}$$

## Statistical significance of trends
In the analysis of tide gauge data and the MITgcm simulations, we estimate the formal standard error of the $M_2$ amplitude (or in-phase and quadrature) trends by evaluating the stochastic model of a least-squares fit. The functional model of the fit consists of a mean value, a linear rate, and a lag one-year autocorrelation of the residual terms, as common in, e.g., studies of mean sea level variability[92]. Allowing for lag-one autocorrelation reduces the degrees of freedom of a full 1993–2020 time series of $M_2$ harmonic coefficients from 28 to typically 18. Corresponding $t$-values for two-sided tests at $\alpha$-levels of 0.05 and 0.32 are 2.101 and 1.023, but slightly higher for lower degrees of freedom (e.g., 2.228 and 1.046 for a gappy tide gauge time series with an effective sample size of 10). We multiply the formal standard error by these location-dependent $t$-values to estimate the 95% and 68% confidence intervals. Standard errors of derived estimates (e.g., sum of trends from two different models in Figs. 5 and 6, or regional averages in Fig. 7) are computed by variance propagation. All uncertainty quantities given in the text and in Table 1 represent 68% confidence limits.

In the altimetry analysis, we account for serial correlation by counting (in each bin) only individual satellite passes as independent data. The resulting formal standard errors for $M_2$ trend components in deep water are $\sim$ 0.06 mm yr$^{-1}$ and several times larger in shallow water (see Supplementary Fig. 4b). To propagate these errors to the smoothed altimetry solutions (Fig. 3a, Supplementary Figs. 6a and 7a), we smooth them with the same boxcar filter as the actual trend estimates and scale the result at any location by $\sqrt{n_1/n_2}$, where $n_1$ represents the number of altimeter tracks crossing the original analysis bin, and $n_2$ is the number of tracks crossing the wider filter bin. This scaling again follows the logic that the number of independent altimetry estimates is approximately equal to the number of crossing tracks. In deep water, we typically have $n_1 = 4$ and $n_2 = 16$, leading to halving of the original formal error. Given that the effective sample size (in time) at any altimetry grid point is difficult to determine, we simply assume that the 68% (95%) confidence interval corresponds to the onefold (twofold) standard error.

## Area averages and associated uncertainties
Trends listed in Table 1 under column "3D model" are linear rates fitted to the simulated, area-averaged $M_2$ amplitude anomalies (1993–2020) depicted in Supplementary Fig. 5. We handle the stochastic component of the fit, including the computation of confidence intervals, as in the preceding section. For the altimetry, which is not a time series, we form a weighted average of the unsmoothed gridded trends (Supplementary Fig. 4a) in each region, with weights set to the inverse of the standard error squared (Supplementary Fig. 4b). The formal error of the resulting trend is estimated as the mean standard error over the considered domain, scaled by $\sqrt{1/a}$. Here, $a$ is a conservative count of non-overlapping (i.e., uncorrelated) deep-ocean bins ($1.5° \times 6°$) within each region.

## Data availability
Grid files of barotropic and baroclinic $M_2$ amplitude, in-phase, and quadrature trends (1993–2020) from the MITgcm simulations have been placed

at https://doi.org/10.5281/zenodo.10844368[93]. All other datasets used in this study are available from the following links: GESLA-3 tide gauge records (https://www.gesla.org/[86–88]), TPXO9 tidal atlas (https://www.tpxo.net/global/tpxo9-atlas[64]), T/P-Jason altimetry (http://rads.tudelft.nl/rads/rads.shtml[79]), C3S gridded sea level anomalies (https://doi.org/10.24381/cds.4c328c78[77]), empirical baroclinic tides (https://ingria.ceoas.oregonstate.edu/~zarone/downloads.html[32]), GLORYS12 Version 1 monthly mean fields (https://doi.org/10.48670/moi-00021[28]), ICE-6G_C crustal displacement (https://www.atmosp.physics.utoronto.ca/~peltier/data.php[76]), and RTopo-2 global bathymetry (https://doi.org/10.1594/PANGAEA.856844[61]).

## Code availability

At https://doi.org/10.5281/zenodo.10844368[93], we provide pre- and post-processing scripts, the MITgcm LLC1080 namelist, and input files for users to repeat the modeling experiment for the year 2006. The barotropic model code, originally adapted from https://geo.mff.cuni.cz/~einspigel/debot.html, is available from M.S. upon request.

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

## Acknowledgements
This work was supported by the German Research Foundation (DFG, Project nos. 451039647 and 388296632). R.D.R. was supported by the U.S. National Aeronautics and Space Administration through the Ocean Surface Topography project. We gratefully acknowledge the Gauss Centre for Supercomputing e.V. (www.gauss-centre.eu) for funding this project by providing computing time through the John von Neumann Institute for Computing (NIC) on the GCS Supercomputer JUWELS at Jülich Supercomputing Centre (JSC). We also acknowledge the granted access to the Bonna cluster hosted by the University of Bonn. Help by Dimitris Menemenlis and Hong Zhang in setting up the LLC1080 simulations is greatly appreciated.

## Author contributions
M.S. conceived the study and interpreted the results. L.O. carried out the model simulations, performed most of the analysis, created figures, and wrote the initial manuscript draft, which was later fleshed out by M.S. R.D.R. contributed with the altimetry analysis, including its textual description, and a guide to compute global dissipation rates. All authors reviewed and commented on the manuscript at multiple stages.

## Funding

## Competing interests
The authors declare no competing interests.
