## [Peer Review File · Communications Earth & Environment]

15th Dec 23

Dear Ms Opel,

First of all, please allow us to apologise for the delay in sending a decision on your manuscript titled "A likely role for stratification in secular changes of the global ocean tides". It has now been seen by two reviewers, and we include their comments at the end of this message. They find your work of interest, but some important points are raised. We are interested in the possibility of publishing your study in Communications Earth & Environment, but would like to consider your responses to these concerns and assess a revised manuscript before we make a final decision on publication.

We therefore invite you to revise and resubmit your manuscript, along with a point-by-point response that takes into account the points raised.

In particular, we will need you to address in depth the requests for an assessment of statistical significance, and a more detailed explanation and justification of the model set-up. Please also ensure that the methodology is explained at a level of detail that will allow an independent researcher to replicate your results (please note, there is no length limit in the Method section).

Please highlight all changes in the manuscript text file.

Please use the following link to submit your revised manuscript, point-by-point response to the referees' comments (which should be in a separate document to any cover letter), a tracked-changes version of the manuscript (as a PDF file) and the completed checklist:

[link redacted]

We hope to receive your revised paper within six weeks; please let us know if you aren't able to submit it within this time so that we can discuss how best to proceed. If we don't hear from you, and the revision process takes significantly longer, we may close your file. In this event, we will still be happy to

reconsider your paper at a later date, as long as nothing similar has been accepted for publication at Communications Earth & Environment or published elsewhere in the meantime.

Please do not hesitate to contact us if you have any questions or would like to discuss these revisions further. We look forward to seeing the revised manuscript and thank you for the opportunity to review your work.

Best regards,

Jose Luis Iriarte Machuca, PhD

Editorial Board Member

Communications Earth & Environment

Heike Langenberg, PhD

Chief Editor

Communications Earth & Environment

EDITORIAL POLICIES AND FORMATTING

Editorial Policy: Policy requirements (Download the link to your computer as a PDF.)

Furthermore, please align your manuscript with our format requirements, which are summarized on the following checklist:

Communications Earth & Environment formatting checklist

and also in our style and formatting guide Communications Earth & Environment formatting guide .

*** DATA: Communications Earth & Environment endorses the principles of the Enabling FAIR data project (<http://www.copdess.org/enabling-fair-data-project/>). We ask authors to make the data that support their conclusions available in permanent, publically accessible data repositories. (Please contact the editor if you are unable to make your data available).

All Communications Earth & Environment manuscripts must include a section titled "Data Availability" at the end of the Methods section or main text (if no Methods). More information on this policy, is available at <http://www.nature.com/authors/policies/data/data-availability-statements-data-citations.pdf>.

If a community resource is unavailable, data can be submitted to generalist repositories such as figshare or Dryad Digital Repository. Please provide a unique identifier for the data (for example a DOI or a permanent URL) in the data availability statement, if possible. If the repository does not provide identifiers, we encourage authors to supply the search terms that will return the data. For data that have been obtained from publically available sources, please provide a URL and the specific data product name in the data availability statement. Data with a DOI should be further cited in the methods reference section.

REVIEWER COMMENTS:

Reviewer #2 (Remarks to the Author):

The work of Opel et al. named "A likely role for stratification in secular changes of the global ocean tides" focuses on the changes observed in the barotropic tide using a set of 3D global simulations that resolve the internal tides generation and propagation. Using a yearly mean stratification for each simulation, they can build the time series of main tidal constituent amplitude and analyze the changes with root mean square and trends calculations. Then, they compare those results of these tidal constituents to the trends of tidal amplitude (1) due to the sea level rise using a similar set of barotropic tidal model with different yearly sea level, (2) from altimetry with yearly tidal analysis and (3) from tides gauges with yearly tidal analysis. Their major result is that the changes in tidal amplitude due to the stratification are obviously linked to the barotropic-to baroclinic tidal conversion (as expected) but above all that those changes are the same order of magnitude, even stronger in open ocean and some shelves, than the changes due to the sea level. I acknowledge the effort that has been done in this study and think that it would be a good contribution for the scientific community.

However, I struggled to understand some methods that have been used and in particular how the authors explained, showed and discussed the significance of the trends they calculated. Actually in this manuscript, only the trends of MITgcm have significance mentioned (confidence interval contours in fig. 3b, 4, 5, 6) whereas I expected to see them for the barotropic model, the altimetry and the tides gauges. In addition, the method section named "Statistical significance of trends" is, in my opinion, too brief and does not explain how multiple single trend's confidence intervals can lead to geo-referenced confidence interval contours for all the trends as shown in the figures. Finally, the choice made to display those confidence interval contours misleads the reading of the figure as I cannot know if the values are inside or outside the contours.

In the present version of the manuscript I do not know how to evaluate the significance of the results, thus I believe that a major modification is needed before I can fully assess this work in a second round review. I also list some more detailed comments below, sorted following the manuscript order.

Simon Barbot

MAJOR COMMENTS / REMARKS:

Fig 1/3/4/5/6:

The white color should not be in the colorbar (the white color does not refer to trends $< -30 \text{ J.m}^{-3}$). For a better clarity I suggest to hatch the non-significant trends rather than showing the contours/white mask.

Fig. 3 : As the 95% confidence interval of the trends have been done on MITgcm, this should also be done for altimetry and barotropic model trends .

I.116: the fact that trends of Fig.3a are “primarily” negative is not clear (however quite clear in Fig.3b). Maybe you should add the mean trend or the distribution of the trends (significant trends only) to strengthen your affirmation.

I.120: I agree that there is a good agreement in the Mozambique channel but there is none in the southern Indian Ocean.

I.128-130: Unclear sentence. Yes a stronger stratification causes a strong IT generation and thus an decrease of M2 barotropic tides (negative trends in 3b). Some areas are taken as examples of such trend linkage but Mascarene ridge, Luzon strait and West pacific have both negative and positive trends... Maybe this is due to the ambiguity of the 95% confidence interval contours (we do not know if a location is inside or outside the confidence interval), and might be resolved with hatching as suggested above.

Figure 5/6: Are all the trends of tide gauges significant ? If not, please add a cross on top of the colored dot.

I.188-191: you discuss the disparity of the Gulf of Alaska compared to the tide gauges even if they both show negative trends whereas you do not discuss the opposite trends that cans be observed in East of Australia, South of Japan, almost half of European tide gauges and central East coast of USA.

Figure 7: The standard errors shown in this figure refers to the distribution of the trends in each region, right ? Have you discarded the trends that were non-significant ?

I.192-195: If the trends are not significant, the average of insignificant trends cannot become significant...

I.297 (Equation 4): you cannot subtract the surface elevation with bottom pressure.

η (the surface expression of the modeled internal tide / surface elevation \rightarrow meter ??)

p_b (bottom pressure \rightarrow Pa)

So gravitational acceleration (g) and water density might be needed for this conversion, please verify it.

MINOR COMMENTS / REMARKS:

Fig1: Replace “Colored markers” by “Yellow dots”

Some letters are difficult to read (A,B,I,H), bigger font and white contours (or white facecolor with black contours) should ease the reading.

l.75: “Additional variability occurs as a result of modulations in source strength,” and wavelengths as well. The citation you used also highlights that the ratio between the modes is changed with the stratification variability. Please rephrase for clarity.

l.78-84: You seem to discard the variability of the ITs generated near the Gulf of Maine because the variability might be mainly due to the interaction with mesoscale eddies and circulation. This is also true for the internal tide generation north of Brazil and along the Kuroshio.

Additional references for the North Brazilian Current:

Tchilimbou et al. (2022) <https://doi.org/10.5194/os-18-1591-2022>

Additional references for the Kuroshio:

Chen et al. (2022) <https://doi.org/10.1016/j.pocean.2022.102863>

Additional references for the Gulf Stream:

Duda et al. (2018) <https://doi.org/10.1175/JPO-D-18-0031.1>

Additional references of academical modeling on eddy/current interactions with ITs:

Dunphy and Lamb (2014) <https://doi.org/10.1002/2013JC009293>

Ponte and Klein (2015) <https://doi.org/10.1002/2014GL062583>

Dunphy et al. (2017) <https://doi.org/10.1175/JPO-D-16-0099.1>

Ponte et al. (2017) <https://doi.org/10.1002/2016JC012214>

l.84-87: The total RMS plot (~non-phaselock ITs) does complete the empirical mapping of the ITs (phaselock) but it should be more relevant to compare it to the studies that investigate the non-phaselock ITs.

Additional references for the ITs empirical mapping (phaselock)

Ubelmann et al. (2022) <https://doi.org/10.5194/os-18-469-2022>

Additional references for non-phaselock ITs estimation:

Zaron (2017) <https://doi.org/10.1002/2016JC012487>

Buijsman et al. (2018) <https://doi.org/10.1002/2016JC012590>

I.90-93: List areas with order of 5cm before the ones with order of 1-3 cm.

I.145-147: I do not understand what anomaly you are referring to. Is it the trends of figure 3a or 3c ?

Figure 4: It shows that the ITs amplitude is increasing worldwide, but interestingly only two areas show negative trends (off Nova Scotia, Canada and off North of Sumatra). You do not discuss this anywhere. Is it caused by the weakening of stratification ?

I.192-204: In my opinion, Figure 6 should be discussed with figure 5 and figure 7 should be discussed afterward.

I.220: “enhanced barotropic-to-baroclinic tidal conversion over steep topography.”

Methods MITgcm

Is $1/12^\circ$ a sufficient resolution for Indonesian seas (with very tight straits but with a strong IT generation) ? The resolution might also be a problem for barotropic tide propagation in the North sea and Denmark straight and might explain the disagreements between MITgcm and tides gauges (Fig. 5b). This is very lightly discussed in this study.

nudging (I.269-270):

Studying gravity waves while using a nudging scheme in order to maintain the stratification is quite tricky as the nudging provokes unwanted residual wave signatures at the relaxation frequency that could affect the original gravity wave signal.

Some studies propose to use a spectral nudging scheme that introduces a frequency threshold to restrain the nudging on low-frequency only. This methods is more common for atmospheric study but here is two references that use this scheme for internal tides:

Kodaira et al. 2019 (<https://doi.org/10.1115/OMAE2019-95842>)

Barbot et al. 2022 (<https://doi.org/10.1029/2022JC018816>)

I rapidly looked at the rbc package documentation (https://mitgcm.readthedocs.io/en/latest/phys_pkgs/rbc.html) but did not find anything similar. Have you done some sensitivity tests to quantify the erosion of the internal tides on your simulations ? This might only have a minor impact close to the generation zone but it should have a stronger impact far from it.

Data availability :

For GLORYS12, use the DOI : <https://doi.org/10.48670/moi-00021>

Reviewer #3 (Remarks to the Author):

Review of “A likely role for stratification in secular changes of the global ocean tides”

Assessment of the various aspects of the manuscript, asked by Communications Earth & Environment:

1. Key results: the role of the stratification in long-term changes of M2 tide is discussed
2. Validity: Does the manuscript have flaws which should prohibit its publication? No.
3. Originality and significance: the conclusions are original and significant. The results presented are of immediate interest to many people in the discipline related to sea level changes.
4. Data & methodology: Data are well presented. Some improvements could be undertaken to clarify the Methods (stratification vs sea level effect, 3D vs 2D model, see the detailed comments).
5. Appropriate use of statistics and treatment of uncertainties: The use of statistics is appropriate. Only one main comment: it is not clear if all the trends plotted are significant at 95% (for example). If they are not, this should be highlighted in the figures (see the detailed comments).
6. Conclusions: the conclusions and data interpretation are robust, even if the results could be interpreted more largely, in particular for the comparison between observed and modeled M2 trends. Large discrepancies exist, showing that in some regions, some important drivers are still missing (in 5 among 13 regions, less than 50% of M2 trends is captured). See the detailed comments below.
7. Suggested improvements: see the detailed comments below, that hopefully will help strengthening the work in a revision.
8. References: Does this manuscript reference previous literature appropriately? Yes.

9. Clarity and context: The paper is well written. Is the abstract clear, accessible? Yes. Are abstract, introduction and conclusions appropriate? Yes. In addition, standard errors associated to the trends could be specified in the Abstract, and a larger discussion would be welcome in the data interpretation.

10. Inflammatory material: Does the manuscript contain any language that is inappropriate or potentially libelous? Not at all.

Abstract:

The present paper focuses on the likely role of stratification in long-term changes of the M2 tide. A 3D global model at $1/12^\circ$, forced only with astronomical tide (M2, S2, K1 and O1) and yearly varying density structures, is implemented. Increased stratification over 1993-2019 causes trends on barotropic M2 tide, of order 0.1 mm/yr in the open ocean, predominantly negative, and up to 0.4 mm/yr in coastal areas. Such trends in the open ocean are due to enhanced energy transfer to internal tides since the 1990s. Comparison between M2 trends due to stratification (present study) and sea level rise (update of a previous study) suggests a dominant role of stratification. Finally, adding these two contributions (stratification and mean sea level rise), the resulted M2 modeled trends are compared with observed trends at tide gauges.

General comments:

The paper tackles the question of drivers of tidal changes over 27 years, focusing on the role of stratification. The paper argues that this process is dominant compared to others, such as the sea level rise. The paper is well written, with many interesting and novel results, and really worth being published in *Communications Earth & Environment*, as it is a great contribution to the question of long-term tidal changes. However, the comparison between modeled and observed M2 trends show strong discrepancies in some regions, which should be underlined in a larger discussion. This gives really place to other drivers. Moreover, the separation between “stratification effect” and “sea level rise effect” needs to be clarified in the paper, as both effects are partially linked (i.e. through steric effects). Our major comments are the following, we hope they will help to strengthen the paper:

1) The term “secular” is used whereas the paper covers only a short period 1993-2019 (27 years). As the paper focus on trends, and knowing the strong interannual to multidecadal variability in sea level data (IPCC, 2021), “long-term” changes would be more appropriate.

2) As soon as the models, with stratification effect and sea level rise contribution, simulate less than 50 % of the observed M2 trends in many of the selected coastal regions (5 regions among 13, see Fig. 7), it means that in these regions either the models do not correctly simulate the undergoing processes, either we are missing some other significant drivers. The same way, the comparison between modeled and observed M2 trends (Fig. 5 and 6) should be analyzed in a more quantitative way (at least, giving the mean correlation or a plot between observed and modeled M2 trends). We suggest some additional diagnostic to estimate more quantitatively how much of the M2 observed trends (in %) is well captured by the model, and where. The mean sea level alone can not explain changes in the open ocean, and the

paper demonstrates clearly the role of stratification in the zones of rough topography and generation of internal waves. In coastal areas, strong discrepancies appear, and could be discussed more deeply, giving some place to other processes, that may not be already investigated (in the literature).

3) The “stratification” effect and “sea level rise” effects are separated, whereas changes in stratification contribute to changes in mean sea level (steric effect). We understand that this part has been corrected, but it should be discussed earlier in the paper. Moreover, this aspect could be clarified, as the Boussinesq hypothesis suggests that there is no steric effect in the model (see the comments further, in more details).

4) Are the trends plotted (for example Fig. 3) all significant? (at 95% for example)

5) The sensitivity for the stratification is conducted with a 3D model, and sea level rise with a 2D shallow-water model, which is not fully consistent. The 3D model could be used to investigate the sea level rise effect (or at least, it could be tested with 1 year of simulation, to check that the results are similar with the 2D/3D model).

6) There is no illustration of time series (yearly stratification/M2, yearly modeled/observed M2), only estimated “linear” trends, whereas it is not evident that these changes are “linear”. Some time series at few points could help to understand the discrepancies between the models and the observations.

In more details (most of those comments are minor):

- line 1: “secular changes”, “long-term” would be more appropriate than “secular” in the title (and in the paper), as the paper focuses on a recent short period of 27 years (1993-2019).

- line 2 : “ocean tides”, “M2 ocean tide” would be more appropriate, as the paper focuses mainly on M2 (S2, K1 and O1 being only evoked in the concluding remarks, and Supplementary)

- line 8: in the abstract, please give associated standard errors for the trends (0.1 mm/yr and 0.4 mm/yr)

- line 12: “typically ~1–3 cm in amplitude”, please precise approximately on which period (over a century?)

- line 16: “the broader oceanic” please add “or atmospheric factors”, the possible role of atmospheric circulation on the tide being also explored by the scientific community (even if not mentioned in this paper). Müller (2011) suggested a link between M2 changes and atmospheric dynamics in the North Atlantic. Pan et al. (2019) suggested that changes in the response of the nodal modulation of the M2 tide from 1970s to 2013 may be linked with the NAO. More recently, Challis et al. (2023) investigated atmospheric wind and pressure-driven changes in tidal characteristics over the Northwestern European Shelf.

- line 16: “However, most regional assessments (see references above) point to a certain degree of spatial coherence in the patterns of tidal change, likely bearing on the role of broader oceanic, rather than anthropogenic factors.” This could be reformulated, as oceanic changes may also have anthropogenic origin (i.e. sea level rise). Maybe the following would be more appropriated “rather than local factors, such as human-induced coastal development”.

- line 21: the importance of investigating tidal changes for “projections of extreme sea levels” could be moderated (see also lines 45-47, lines 236-237), even if we agree that many authors use this argument. Changes in extreme sea levels are firstly driven by sea level rise (globally ~ 3.1 mm/yr since the 1990s, Dangendorf et al. (2017)), whereas changes in the tide are from far second order: ~ 0.1 mm/yr (see Fig. 3), which means around 30 times smaller.

- lines 25-26: “However, at most locations, especially those open to the sea, sea level rise alone is insufficient to explain the observed trends in tides (Schindelegger et al., 2018). Could changes in ocean stratification be a more relevant forcing factor?” It is not clear here why jumping from mean sea level rise as a major driver (which was a strong argument in Schindelegger et al. (2018), arguing that the answer is “Yes, or mostly” to the question “Can We Model the Effect of Observed Sea Level Rise on Tides?”) to stratification as a possible dominant driver (the present paper). We suggest here to give more details from the previous paper, to give consistency between the two papers. Despite Schindelegger et al. (2018) found that the sign of the M2 trends were well reproduced, their amplitude were really weaker than observations, “within a factor of 4 (or less) from each other in nearly 50% of the considered cases”, suggesting that other important drivers were missing (among them, the stratification being a candidate). In fact, both sea level rise and stratification probably acts (and possibly others), and which one is dominant mainly depends from each region (e.g. stratification being dominant in regions of rough topography, where internal waves are generated).

- line 39-47 “Our simulations suggest...sea levels.”, these are results that should rather be in the following section Results and discussion, or in Abstract/Concluding remarks?

- line 49 : “Basic assessment” could be “Basic assessment of the model”, to be more precise

- line 52: “27 annual simulations” instead of “27 simulations” would be clearer

- line 62 : “see Supplementary Fig. 1 and Fig. 1 (black squares)” perhaps rather “see black squares on Fig.1 and also Supplementary Fig. 1” (otherwise “Fig. 1 and Fig. 1” is confusing)

- line 70 “M2 sensitivity to stratification changes” rather “Sensitivity to stratification changes”

- line 73-74 : “The picture is evidently dominated by temporal variability of the baroclinic tide, primarily reflecting the impact of stratification changes on wave propagation characteristics^{34, 35}. A general comment for this paragraph: would it be possible to clarify how the Fig. 2 can be interpreted as sensitivity to “baroclinic” tide (lines 73-87) and “barotropic” tide (lines 88-102), whereas both are mixed on Fig. 2? We guess that when strong M2 variability occur in regions of generation of internal tides (areas of rough topography, black squared on Fig. 1), it is the signature of baroclinic tide. To highlight this, black dots (similarly to black squares on Fig. 1) could be added on Fig. 2. Elsewhere, we understand that M2 variability is mainly the signature of M2 barotropic tide, is that right?

- line 103: “Long-term M2 trends – global synthesis” rather than “Secular M2 trends - global synthesis”

- line 106: “2D shallow-water model”, the sensitivity for the stratification is conducted with a 3D model, and sea level rise with a 2D shallow-water model, which is not fully consistent. Moreover, using the 2D shallow-water model to get rid of the sea level effect for the 3D stratification model also add complexity in the analysis (see below comment lines 333-344). We suggest to run the 3D model to investigate the sea level rise effect (or demonstrate with 1 year of simulation, that the results are similar). All the

modeling part could be conducted with the 3D model and two experiments 1) an annually varying density structures simulation (correcting carefully outputs from the steric effect, if needed), as in the paper, and 2) an annually changing bathymetry simulation, to reproduce 27 years of sea level rise.

- Fig. 3 (a): the standard errors are quite large for M2 trends (~ 0.05 mm/yr, Supplementary Fig. 4). Is it possible to plot only the significant M2 trends on Fig. 3 (a)? (at 95% for example, not significant trends being for example in white, or hatched).

- Fig. 3 (b) and (c): same comment, are all the trends significant considering the autocorrelation in the data? (See the statistical significance of trends in Methods). If not, only significant trends at 95% (for example) should be plotted (non significant trends being for example in white, or hatched).

- Fig. 3: in the following of the paper, (b) and (c) are added to be compared to tide gauges trends. The same way, to be consistent, could (b) and (c) be added to be compared with (a)?

- lines 126-127: "Fig. 3 shows that there are indeed contemporary large-scale trends of M2 in the open ocean and that these signals are primarily caused by changes in ocean stratification rather than by relative sea level rise", to demonstrate it quantitatively rather than qualitatively, would it be possible to conduct some additional diagnostics: for example, in %, how much of the observed trends (a) are captured by (b) and (c)?

- line 128: "The general tendency toward weaker M2 amplitudes", again, would it be possible to be quantitative? What is the global mean M2 trend, in Fig. 3 (a), (b) and (c)?

- lines 155-157: "we again find reasonable agreement between M2 amplitude trends in the baroclinic model and those inferred from satellite altimetry" again, would it be more quantitative, with for example a global correlation map between M2 modeled time series and M2 observed time series over 1993-2019? (whose trends are Fig. 3b and 3a) Or showing what is the % of (b) compared to (a)? (i.e. how much of the observed trends are captured by the model). For example, for the "Northwest European Shelf" where there is a "reasonable agreement" (line 157), the observed trends seems very small (~ 0.05 mm/yr), compared to the modeled ones (~ 0.15 mm/yr). The % of captured trends would really help to see where the stratification process is a dominant driver.

- lines 165-166: "We now shift the focus toward the coast and compare our simulation results against measured M2 amplitude trends at tide gauges." As trends are probably not linear, is it possible to have at least a comparison of yearly time series at some tide gauges? (for example, in Supplementary)

- line 176: "By and large, tide gauges and model results point to the same regional patterns of coastal M2 trends since 1993", again, would it be possible to be more quantitative? Giving for example a plot 'modeled/observed trends', or at least the correlation by region between modeled and observed trends. It seems that some regions are better modeled than others, and large discrepancies appear for example in Fig. 5b. This could be underlined, opening the discussion to possible other drivers.

- line 181: "we form a "budget" of contemporary M2 amplitude trends in 13 selected coastal regions". As shown on Fig.5, some tide gauges are well modeled, whereas others are not. We can suspect some local effects (e.g. coastal development) in tide gauges not correctly modeled. For this reason, the comparison by tide gauge may be more pertinent, than a regional budget.

- line 214: “several, possibly competing processes might be at work”, yes, this may be developed more deeply in the paper. In the introduction, it is mentioned that “However, at most locations, especially those open to the sea, sea level rise alone is insufficient to explain the observed trends in tides”. The same way, in the coastal waters, Fig. 7 shows the dominant role of stratification in some regions, but in many regions (at least 5 among the 13, where less than 50% of M2 trend is captured), stratification and sea level rise can not explain alone the M2 trends.

- line 219: “the primary mechanism”, as a secondary mechanism is not further discussed, we suggest the “main mechanism”?

- line 237: “which presently ignore changes in tides or consider them as a function of sea level rise alone”, please reformulate. The changes in tide are not always ignored. As soon as the projections are conducted with a 3D model taking into account the stratification, stratification-induced changes in tide will also be considered. If projections are based on a barotropic model, changes in tide due to stratification will effectively be ignored (but those due to mean sea level will be considered). Again, changes in tide (~ 0.1 mm/yr) are second order compared to changes in mean sea level (~ 3 mm/yr).

- lines 239-277: In the “Modeling approach”, the forcing should be more explicitly mentioned. We understand that there is only the tidal forcing (earth tidal potential?) with 4 components (M2, S2, K1, O1) and associated SAL tide, and no atmospheric forcing at all (no winds and atmospheric pressure). Could this be expressed more explicitly? In particular, the line 255 is confusing “The only forcing applied is barometric pressure loading...”. We guess this is done externally for SAL tide, and not for the simulations (otherwise, M2 trends could be also due to atmospheric forcing).

- line 326: “26 years”, 27 years?

- line 331: “26 years”, 27 years?

- lines 333-344: this paragraph on the sea level effect linked to changes in stratification is essential and should be discussed earlier in the paper. Otherwise, from the beginning of the paper, it is not clear how to compare “stratification” and “sea level rise” whereas changes in density profiles imply changes in sea level. Concerning this paragraph, we have two major comments:

1) The model being under the Boussinesq approximation, the volumes are conserved. With this hypothesis, how the steric effect could be modeled? It is mentioned line 335: “Differences in η_0 from run to run generally reflect varying amount of steric expansion”. Is the steric effect derived a posteriori from the model temperature (Griffies and Greatbach, 2012)? This aspect should be clarified.

2) To disentangle the steric effect from the stratification effect itself, why not simply consider the water levels (from the 3D model) corrected from the steric effect, and analyse directly M2 trends of these steric-corrected water levels? It is more direct than correcting the total M2 trends (from the 3D model), from the “residual trend signal”, computed using a barotropic model with a perturbed bathymetry to take into account the steric expansion (roughly approximated with trends from the 3D model sea levels).

- line 370: considering the quite large standard deviation for the satellite (0.05 mm/yr), are all the trends from Fig. 3 (a) significant at 95%? If not, only significant trends should be plotted (white or hatched areas for not significant trends).

- line 390: GESLA-3 suggests to cite also Caldwell et al. (2015) and Woodworth et al. (2017), in addition to Haigh et al. (2022), see <https://gesla787883612.wordpress.com/downloads/>

- lines 413-421: again, are all the trends from Fig. 3b and Fig. 3c statistically significant at 95%? If not, only significant trends should be plotted (white or hatched areas).

Dear Reviewers,

We thank you for evaluating our manuscript and appreciate your comments and suggestions. These helped strengthen the manuscript in several places, contributed to a more balanced discussion, and prevented us from drawing incorrect conclusions in one case. We start our letter with a few general remarks and a summary of the main modifications made to the manuscript:

- The new **analysis period is 1993–2020**, consistently implemented across all components of the study.
- We have followed the recommendation by Reviewer #3 and quantified the secular M_2 change in the 3D model due to steric expansion using the very same numerical model (consistency issue). This led to the insight that the **large negative trends in the Irish/Celtic Seas and English Channel (~ -0.4 mm/yr) reported in the initial submission were mainly due to changes in local water column thickness**, and not stratification. Our estimates for this effect also changed in a few other marginal regions of the North Atlantic, but not to an extent that our conclusions would be compromised. We have considered this somewhat unfavorable sensitivity in our discussion.
- Three coastal regions with low-magnitude and/or statistically insignificant M_2 trends have been removed from Figure 7 in the “Regional foci” section (Japan, Gulf of Mexico West, Gulf of Maine). Moreover, a fresh inspection of the tide gauge time series showed some anomalous annual estimates (e.g., at Liverpool and Milford) or overall questionable records (Crookhaven, AUS). Elimination of the suspicious data and addition of the year 2020 moderately changed the observed trends in Figure 7.
- We have worked on the assessment of statistical significance and the graphical display of this information. In particular, in our central Figure 3, we are now showing a **spatially smoothed version of the altimetric M_2 trend** (with stippling for confidence intervals) and a revised trend map from the barotropic model (also with uncertainty information). For the latter, we have gone all out and done annual time slice simulations, replacing the two endpoint simulations of the initial submission.
- Our original description of the model setup was already fairly comprehensive, but we agree that some elements (e.g., why $1/12^\circ$ resolution, limitations of the model and nudging scheme) along with a few important hints for readers (“local steric effects”) were missing. This has been rectified. In this context, note that a $1/12^\circ$ 3D setup for multiple global tide simulations is the best one can do at the moment. The next higher LLC configuration of the MITgcm is $1/24^\circ$ (~ 8 Mio core-h, 700 TB 3D output) and may be feasible in a few years from now, but it likely won’t solve all problems.
- We would like readers to **adopt realistic expectations as to how well observed and simulated M_2 trend can agree**. Extracting such trends from models and data (especially satellite altimetry with all its aliasing issues!) is quite a different beast compared to, e.g., studies of regional sea level rise, where effects are much larger and pointing all in the same direction. For tidal trends, it is not long ago that the news was about getting the sign right (see the Schindelegger et al. 2018). An agreement within a factor of 2 or 3 actually marks progress—no global study has ever come that close, and the comparatively good correspondence we are showing lends further credence to our modeling approach.
- We would also like to encourage readers to not over-rely on binary determinants—such as confidence limits from (always approximate) error considerations—to judge the significance of our (or any) results. Coherent regional patterns within a dataset and

correspondence between independent estimates of the same effect can also be a measure of significance. No perfect statistical approach can leverage this capacity. A good example in this regard is Figure 3a vs. 3b in, e.g., the Indian or East Pacific Ocean, or the M_2 in-phase component (Supplementary Figure 6) for most of the global ocean.

(Comments received on 15 December 2023)

Reviewer #2:

(Line numbers in our responses refer to the PDF of the revised manuscript.)

The work of Opel et al. named “A likely role for stratification in secular changes of the global ocean tides” focuses on the changes observed in the barotropic tide using a set of 3D global simulations that resolve the internal tides generation and propagation. Using a yearly mean stratification for each simulation, they can build the time series of main tidal constituent amplitude and analyze the changes with root mean square and trends calculations. Then, they compare those results of these tidal constituents to the trends of tidal amplitude (1) due to the sea level rise using a similar set of barotropic tidal model with different yearly sea level, (2) from altimetry with yearly tidal analysis and (3) from tides gauges with yearly tidal analysis. Their major result is that the changes in tidal amplitude due to the stratification are obviously linked to the barotropic-to baroclinic tidal conversion (as expected) but above all that those changes are the same order of magnitude, even stronger in open ocean and some shelves, than the changes due to the sea level. I acknowledge the effort that has been done in this study and think that it would be a good contribution for the scientific community.

However, I struggled to understand some methods that have been used and in particular how the authors explained, showed and discussed the significance of the trends they calculated. Actually in this manuscript, only the trends of MITgcm have significance mentioned (confidence interval contours in fig. 3b, 4, 5, 6) whereas I expected to see them for the barotropic model, the altimetry and the tides gauges. In addition, the method section named “Statistical significance of trends” is, in my opinion, too brief and does not explain how multiple single trend’s confidence intervals can lead to geo-referenced confidence interval contours for all the trends as shown in the figures. Finally, the choice made to display those confidence interval contours misleads the reading of the figure as I cannot know if the values are inside or outside the contours.

In the present version of the manuscript I do not know how to evaluate the significance of the results, thus I believe that a major modification is needed before I can fully assess this work in a second round review. I also list some more detailed comments below, sorted following the manuscript order.

Simon Barbot

Thank you for taking the time to assess our work and for providing many good suggestions. We have made particular efforts to better display (and for the barotropic model to include) confidence bounds on Figures 1, 3, 5, and 6. We have also added a few more details to the “Statistical significance of trends” section, but it is still brief, because there is nothing complicated to write about—we estimate formal standard errors for the various datasets (models, altimetry, tide gauges),

and scale them to confidence intervals as appropriate; if formal errors are required for a derived estimates (e.g., sum or average of trends), they are deduced using the rules of variance propagation. Frequent use of 68% confidence intervals is consistent with our title, as they mark the low end of “likely” probability for a trend outcome.

MAJOR COMMENTS / REMARKS:

Fig 1/3/4/5/6:

The white color should not be in the colorbar (the white color does not refer to trends $< -30 \text{ J.m}^{-3}$). For a better clarity I suggest to hatch the non-significant trends rather than showing the contours/white mask.

Thanks. We have removed the arrow ends of the colorbar in Figure 1. Non-significant trend values are now shown in color, but stippled white for clarity (hatching is difficult to realize with the software we are using).

Fig. 3: As the 95% confidence interval of the trends have been done on MITgcm, this should also be done for altimetry and barotropic model trends.

68% and 95% confidence intervals are now consistently implemented across panels in Figure 3. For altimetry, we had to adopt a smoothing approach to display the uncertainty information in a meaningful way and emphasize the larger spatial scales; the unsmoothed version is included in the Supplement. For barotropic trends, we originally deduced them from two endpoint simulations 26 years of sea level rise apart, giving error-free trends (standard approach in the field). With the annual barotropic time slice simulations performed for this revision, it became possible to estimate formal errors/confidence intervals (the two approaches give nevertheless very similar trend results).

1.116: the fact that trends of Fig.3a are “primarily” negative is not clear (however quite clear in Fig.3b). Maybe you should add the mean trend or the distribution of the trends (significant trends only) to strengthen your affirmation.

With the smoothed map now shown in Figure 3a, the tendency for negative trends is also clear in the altimetry solution.

1.120: I agree that there is a good agreement in the Mozambique channel but there is none in the southern Indian Ocean.

Correct, we now only mention the tropical Indian Ocean.

1.128-130: Unclear sentence. Yes a stronger stratification causes a strong IT generation and thus an decrease of M2 barotropic tides (negative trends in 3b). Some areas are taken as examples of such trend linkage but Mascarene ridge, Luzon strait and West pacific have both negative and positive trends... Maybe this is due to the ambiguity of the 95% confidence interval contours (we do not know if a location is inside or outside the confidence interval), and might be resolved with hatching as suggested above.

We don't think the sentence is unclear; when we write about “general tendency” and “likely” effects, it implies that there can be exceptions from the overall picture. No changes made to the text but to the graphical presentation of the uncertainty information (see above).

Figure 5/6: Are all the trends of tide gauges significant? If not, please add a cross on top of the colored dot.

Studies in cartography have found that fuzzy boundaries for markers are one of the most efficient means for conveying a sense of uncertainty. Here, we can't really have fuzzy boundaries, but we mimic them with the white edge color (black is for trends above the formal error).

1.188-191: you discuss the disparity of the Gulf of Alaska compared to the tide gauges even if they both show negative trends whereas you do not discuss the opposite trends that can be observed in East of Australia, South of Japan, almost half of European tide gauges and central East coast of USA.

The Gulf of Alaska signal is one of the more robust ones (model-altimetry-TGs see trends with similar sign and magnitude), hence the discussion; the Central East coast of USA is in fact also discussed (as "Mid-Atlantic Bight"). Conflicting trends at TGs in Japan and no clear signal in the simulations prevented us from expanding on this region—we have removed the region from Figure 7. Perhaps one could do the same for Southeast Australia, and replace it with the Skagerrak/Kattegat region, but then again, the Skagerrak/Kattegat is not very interesting from a tidal perspective (M_2 amplitude ~ 10 cm). Anyway, even with the rewrite of the coastal section, we cannot go through each and every region—that would be quite tedious. Sometimes it is better to leave secondary results for the reader to discover (in figures and tables).

Figure 7: The standard errors shown in this figure refers to the distribution of the trends in each region, right? Have you discarded the trends that were non-significant?

Yes, in the original version the error bars in Figure 7 were standard errors, as deduced from propagating the standard error (squared = variance) at individual tide gauges to the average. In the revised version, we proceed similarly, but show confidence limits. Locations with insignificant trends are now discarded (but yes, some of them slipped into the average in the original submission). We added all necessary information to the caption of Figure 7.

1.192-195: If the trends are not significant, the average of insignificant trends cannot become significant.

We were confused by our own display of confidence intervals, but in any case, the sentence is now obsolete.

1.297 (Equation 4): you cannot subtract the surface elevation with bottom pressure. η (the surface expression of the modeled internal tide / surface elevation \rightarrow meter ??) p_b (bottom pressure \rightarrow Pa) So gravitational acceleration (g) and water density might be needed for this conversion, please verify it.

Thank you, corrected.

MINOR COMMENTS / REMARKS:

Fig1: Replace "Colored markers" by "Yellow dots". Some letters are difficult to read (A,B,I,H), bigger font and white contours (or white facecolor with black contours) should ease the reading.

OK, "Colored markers" replaced with "Yellow markers". We have also scaled up the fonts of all letters and improved the placement in a few cases. The suggested black/white combination doesn't work so well.

1.75: "Additional variability occurs as a result of modulations in source strength," and wavelengths

as well. The citation you used also highlights that the ratio between the modes is changed with the stratification variability. Please rephrase for clarity.

We had to shorten this passage a bit to incorporate some of your other suggestions. We now summarize all effects in one sentence (“... impact of stratification changes on wave propagation characteristics, modal partitioning, and variations in source strength”), with citations given at the end of this sentence.

1.78-84: You seem to discard the variability of the ITs generated near the Gulf of Maine because the variability might be mainly due to the interaction with mesoscale eddies and circulation. This is also true for the internal tide generation north of Brazil and along the Kuroshio. Additional references for the North Brazilian Current:

Tchilibou et al. (2022) <https://doi.org/10.5194/os-18-1591-2022>

Additional references for the Kuroshio: Chen et al. (2022) <https://doi.org/10.1016/j.pocean.2022.102863>

Additional references for the Gulf Stream:

Duda et al. (2018) <https://doi.org/10.1175/JPO-D-18-0031.1>

Additional references of academical modeling on eddy/current interactions with ITs:

Dunphy and Lamb (2014) <https://doi.org/10.1002/2013JC009293>

Ponte and Klein (2015) <https://doi.org/10.1002/2014GL062583>

Dunphy et al. (2017) <https://doi.org/10.1175/JPO-D-16-0099.1>

Ponte et al. (2017) <https://doi.org/10.1002/2016JC012214>

Thanks for these pointers. We have incorporated two of these references (Duda et al. 2018, Tchilibou et al. 2022) and think this is as far as we can go in our discussion of IT variability without risking a break in the narrative. But we agree that the interaction of ITs with the background circulation is an important and interesting topic in its own right.

1.84-87: The total RMS plot (~non-phaselock ITs) does complete the empirical mapping of the ITs (phaselock) but it should be more relevant to compare it to the studies that investigate the non-phaselock ITs.

Discussing the distinction between phase-locked and non-phase-locked ITs and adding the necessary context for the non-expert reader would be a layer too much for this manuscript, especially as it drives at the barotropic tide. In any case, our sentence implies that the RMS plot can be compared to previous results for non-phase-locked ITs; we already cited Zaron (2017) from your list and now added Buijsman et al. (2017).

Additional references for the ITs empirical mapping (phaselock)

Ubelmann et al. (2022) <https://doi.org/10.5194/os-18-469-2022>

Additional references for non-phaselock ITs estimation:

Zaron (2017) <https://doi.org/10.1002/2016JC012487>

Buijsman et al. (2017) <https://doi.org/10.1002/2016JC012590>

1.90-93: List areas with order of 5cm before the ones with order of 1-3 cm.

The 1-3 cm signals are more widespread, so it is warranted to discuss them first. No changes made.

1.145-147: I do not understand what anomaly you are referring to. Is it the trends of figure 3a or 3c?

We have replaced “anomaly” with “signals in the altimetry map”.

Figure 4: It shows that the ITs amplitude is increasing worldwide, but interestingly only two areas show negative trends (off Nova Scotia, Canada and off North of Sumatra). You do not discuss this anywhere. Is it caused by the weakening of stratification?

We do not know for the Sumatra region (looks like a mix of positive and negative IT trends there), but weaker stratification off the Nova Scotian Shelf and less tidal conversion at the shelf break are consistent with the positive M_2 trend inside the Gulf of Maine (Fig. 3b; see <https://doi.org/10.1029/2022GL101671> for a discussion of this “teleconnection”). We have incorporated some of these thoughts in the revised manuscript (lines 223–225).

1.192-204: In my opinion, Figure 6 should be discussed with figure 5 and figure 7 should be discussed afterward.

These three figures clearly belong together. We don't think we are asking too much from readers when introduced all three in the first two paragraphs of the “Regional foci” section.

1.220: “enhanced barotropic-to-baroclinic tidal conversion over steep topography.”

Done.

METHODS

MITgcm: Is $1/12^\circ$ a sufficient resolution for Indonesian seas (with very tight straits but with a strong IT generation)? The resolution might also be a problem for barotropic tide propagation in the North Sea and Denmark straight and might explain the disagreements between MITgcm and tides gauges (Fig. 5b). This is very lightly discussed in this study.

$1/12^\circ$ grid spacing is the current industry standard in global studies of changes in the barotropic tide, see Schindelegger et al. (2018, [10.1029/2018JC013959](https://doi.org/10.1029/2018JC013959)) or Pickering et al. (2017, [10.1016/j.csr.2017.02.004](https://doi.org/10.1016/j.csr.2017.02.004)), who even went down to $1/8^\circ$. Moreover, we don't see any issues with barotropic tide propagation in the MITgcm for the North Sea and Denmark straights. However, for representation of (low-mode) internal tides and tidal conversion, $1/12^\circ$ is certainly the lowest one should go. While the setup may not be fully adequate for representing complex tidal variability in the Indonesian seas, it is a compromise that allows us to run experiments for a global ocean domain. In future, rather than spending ~8 Mio core-h on the next best global MITgcm LLC setup ($1/24^\circ$), dedicated regional studies with km-scale resolution appear to be sensible targets.

We have included a short justification of the MITgcm's horizontal resolution under “Modeling approach” and incorporated some thoughts regarding ITs in the Indonesian seas in the “Model limitations” section.

Nudging (1.269-270): Studying gravity waves while using a nudging scheme in order to maintain the stratification is quite tricky as the nudging provokes unwanted residual wave signatures at the relaxation frequency that could affect the original gravity wave signal. Some studies propose to use a spectral nudging scheme that introduces a frequency threshold to restrain the nudging on low-frequency only. This method is more common for atmospheric study but here is two references that use this scheme for internal tides: Kodaira et al. 2019 (<https://doi.org/10.1115/OMAE2019-95842>); Barbot et al. 2022 (<https://doi.org/10.1029/2022JC018816>). I rapidly looked at the rbc package documentation (https://mitgcm.readthedocs.io/en/latest/phys_pkgs/rbc.html) but did not find anything similar. Have you done some sensitivity tests to quantify the erosion of the

internal tides on your simulations? This might only have a minor impact close to the generation zone but it should have a stronger impact far from it.

Thank you for these insights. While the MITgcm rbc package is obviously less sophisticated than the nudging schemes in the studies you are pointing us to, the good agreement of our modeled IT trends with altimetry (Zhao 2023, <https://doi.org/10.1029/2023GL105764>) suggests that we are doing the right thing. We have performed some sensitivity experiments in the frame our regional Gulf of Maine study, primarily to select the relaxation time scale and verify that the simulation maintains the background stratification. We did not notice big impacts on internal tides away from generation sites. Our simulations are quite band-limited (4 constituents, 3-day nudging time scale), so that may help.

We have summarized your input in two sentences on lines 301–303 in the Methods section added a “personal communication” note at the end. Is this fine by you?

Data availability: For GLORYS12, use the DOI : <https://doi.org/10.48670/moi-00021>
Done.

Reviewer #3:

General comments:

The paper tackles the question of drivers of tidal changes over 27 years, focusing on the role of stratification. The paper argues that this process is dominant compared to others, such as the sea level rise. The paper is well written, with many interesting and novel results, and really worth being published in *Communications Earth & Environment*, as it is a great contribution to the question of long-term tidal changes. However, the comparison between modeled and observed M2 trends show strong discrepancies in some regions, which should be underlined in a larger discussion. This gives really place to other drivers. Moreover, the separation between “stratification effect” and “sea level rise effect” needs to be clarified in the paper, as both effects are partially linked (i.e. through steric effects). Our major comments are the following, we hope they will help to strengthen the paper:

Thanks a lot for your commending words. Concerning your suggestion for a larger discussion, we want to emphasize that this study focuses on the impact of stratification changes on tides. We also consider sea level rise effects, which are relatively well understood, and highlight a few cases where we have more solid ideas of what other processes might be at work (e.g., land reclamation effects in Yellow Sea). This is as far as we can go in terms of discussion in a cogent, self-contained paper, especially when it is the first one that quantifies the stratification effect on the global barotropic tides. We will not add new references and we cannot digress into speculations and the usual enumeration of causes; that’s what review papers, book chapters, and PhD theses are for.

We understand that the following 6 points summarize your detailed comments further down below. Since we directly respond to those comments, we color your summary statements in grey and only write a very short reply.

1) The term “secular” is used whereas the paper covers only a short period 1993-2019 (27 years).

As the paper focus on trends, and knowing the strong interannual to multidecadal variability in sea level data (IPCC, 2021), “long-term” changes would be more appropriate.

Not considered, see our reasoning below.

2) As soon as the models, with stratification effect and sea level rise contribution, simulate less than 50 % of the observed M2 trends in many of the selected coastal regions (5 regions among 13, see Fig. 7), it means that in these regions either the models do not correctly simulate the undergoing processes, either we are missing some other significant drivers. The same way, the comparison between modeled and observed M2 trends (Fig. 5 and 6) should be analyzed in a more quantitative way (at least, giving the mean correlation or a plot between observed and modeled M2 trends). We suggest some additional diagnostic to estimate more quantitatively how much of the M2 observed trends (in %) is well captured by the model, and where. The mean sea level alone can not explain changes in the open ocean, and the paper demonstrates clearly the role of stratification in the zones of rough topography and generation of internal waves. In coastal areas, strong discrepancies appear, and could be discussed more deeply, giving some place to other processes, that may not be already investigated (in the literature).

See our “up front” comment above. As for the desired quantitative comparison between modeled and measured trends, that is what Figure 7 is about. Correlation plots would emphasize year-to-year variability, which will not be discussed in the present manuscript.

3) The “stratification” effect and “sea level rise” effects are separated, whereas changes in stratification contribute to changes in mean sea level (steric effect). We understand that this part has been corrected, but it should be discussed earlier in the paper. Moreover, this aspect could be clarified, as the Boussinesq hypothesis suggests that there is no steric effect in the model (see the comments further, in more details).

The model still produces spatially varying steric effects. The corresponding correction is clearly something for the Methods section, not the main text.

4) Are the trends plotted (for example Fig. 3) all significant? (at 95% for example)

Addressed with appropriate modifications to the figures.

5) The sensitivity for the stratification is conducted with a 3D model, and sea level rise with a 2D shallow-water model, which is not fully consistent. The 3D model could be used to investigate the sea level rise effect (or at least, it could be tested with 1 year of simulation, to check that the results are similar with the 2D/3D model).

The 3D model is too costly to be used for this purpose and we have more confidence in the 2D model for simulating the tidal response to sea level rise. Suggestion not considered.

6) There is no illustration of time series (yearly stratification/M2, yearly modeled/observed M2), only estimated “linear” trends, whereas it is not evident that these changes are “linear”. Some time series at few points could help to understand the discrepancies between the models and the observations.

We now include time series of regional averaged M₂ amplitude changes from the MITgcm and annual harmonics (1993–2020) at four tide gauges in the Supplement. Other than that, Figure 2 (RMS variability) and confidence intervals on trends give a sense of how “linear” the trends may or may not be.

In more details (most of those comments are minor):

- line 1: “secular changes”, “long-term” would be more appropriate than “secular” in the title (and in the paper), as the paper focuses on a recent short period of 27 years (1993-2019).

We disagree. The term “secular” is much more common in available literature on the subject of contemporary trends in tides; see also papers by the 2nd and 3rd author, as well as Bij de Vaate (2022, doi:10.1029/2022JC018845) for the altimetry time span.

- line 2: “ocean tides”, “M2 ocean tide” would be more appropriate, as the paper focuses mainly on M₂ (S₂, K₁ and O₁ being only evoked in the concluding remarks, and Supplementary)

We disagree again. S₂, K₁, and O₁ are part of the simulations and the results; the fact that we couldn't fit much of the material in the main text doesn't change this. More importantly, our work arguably opens up new avenues for research, which concern other tidal constituents as much as M₂. **An inclusive and forward-looking title, which hints at these implications, is much more appropriate.**

- line 8: in the abstract, please give associated standard errors for the trends (0.1 mm/yr and 0.4 mm/yr)

The quoted -0.1 mm/yr for open-ocean trends is an order of magnitude estimate, saying that the regionally coherent trends are neither -0.01 mm/yr nor -1 mm/yr. Adding uncertainty information to such an estimate makes little sense (apart from being very difficult, given that confidence intervals cover a wide range of values). For the coastal M₂ trends quoted in the revised abstract, we have included the note that those are significant at the 95% confidence level.

- line 12: “typically $\sim 1-3$ cm in amplitude”, please precise approximately on which period (over a century?)

The numbers hold for year-to-year or decadal changes as well as for centuries. For clarity, and given the focus on trends in this work, we have added “over a century”.

- line 16: “the broader oceanic” please add “or atmospheric factors”, the possible role of atmospheric circulation on the tide being also explored by the scientific community (even if not mentioned in this paper). Müller (2011) suggested a link between M₂ changes and atmospheric dynamics in the North Atlantic. Pan et al. (2019) suggested that changes in the response of the nodal modulation of the M₂ tide from 1970s to 2013 may be linked with the NAO. More recently, Challis et al. (2023) investigated atmospheric wind and pressure-driven changes in tidal characteristics over the Northwestern European Shelf.

We have replaced “oceanic factors” with “natural factors” to be more inclusive, see also the title written by Jay and Talke (2020).

- line 16: “However, most regional assessments (see references above) point to a certain degree of spatial coherence in the patterns of tidal change, likely bearing on the role of broader oceanic, rather than anthropogenic factors.” This could be reformulated, as oceanic changes may also have anthropogenic origin (i.e. sea level rise). Maybe the following would be more appropriated “rather than local factors, such as human-induced coastal development”.

The suggested addition would result in an overly long sentence. We therefore chose to write “local anthropogenic factors” and expect that the connection to coastal engineering measures (previous

sentence) is clear. More generally, the term “anthropogenic” is now increasingly used in Earth system research to denote human interventions on regional or local scale, separated from environmental effects (e.g., sea level rise) due to greenhouse gas forcing/climate change.

- line 21: the importance of investigating tidal changes for “projections of extreme sea levels” could be moderated (see also lines 45-47, lines 236-237), even if we agree that many authors use this argument. Changes in extreme sea levels are firstly driven by sea level rise (globally ~ 3.1 mm/yr since the 1990s, Dangendorf et al. (2017)), whereas changes in the tide are from far second order: ~ 0.1 mm/yr (see Fig. 3), which means around 30 times smaller.

We think it is a compelling question. Who knows what tidal conversion and vertical mixing will do to the barotropic tide in a much more stratified ocean by the end of the 21st century? Given the importance of the stratification effect on tides over past decades (~ 0.3 mm/yr in several locations), the subject is worth investigating. Changes of ~ 3 – 5 cm in M_2 by 2100 are possible and that would be as large as the extreme sea level contribution due to storm surges and waves (which are routinely considered in such projections). We slightly moderated the last sentence of the main text but left the two references to projections in the Introduction unchanged, since they just indicate that stratification effects on future tides are presently ignored.

- lines 25-26: “However, at most locations, especially those open to the sea, sea level rise alone is insufficient to explain the observed trends in tides (Schindelegger et al., 2018). Could changes in ocean stratification be a more relevant forcing factor?” It is not clear here why jumping from mean sea level rise as a major driver (which was a strong argument in Schindelegger et al. (2018), arguing that the answer is “Yes, or mostly” to the question “Can We Model the Effect of Observed Sea Level Rise on Tides?”) to stratification as a possible dominant driver (the present paper). We suggest here to give more details from the previous paper, to give consistency between the two papers. Despite Schindelegger et al. (2018) found that the sign of the M_2 trends were well reproduced, their amplitude were really weaker than observations, “within a factor of 4 (or less) from each other in nearly 50% of the considered cases”, suggesting that other important drivers were missing (among them, the stratification being a candidate). In fact, both sea level rise and stratification probably acts (and possibly others), and which one is dominant mainly depends from each region (e.g. stratification being dominant in regions of rough topography, where internal waves are generated).

The argument in Schindelegger et al. (2018) was that we had an accurate tide model with inline SAL and a few other improvements (resolution, reference bathymetry, water depth changes) to reliably quantify the sea level rise effect on tides. Given the partial success in reproducing measured trends, the “yes” or “mostly” answer was included in the conclusions. It should be read as “we can explain larger signal fractions than previous studies, thus pointing to improved modeling capabilities”. In any case, the main point is that we found discrepancies of 3 to 5 back then (see Section 4.2 in the 2018 paper). To align the narratives of the two papers a bit better, we have changed the final sentence to this paragraph to “relative sea level rise can only explain fractions ($\lesssim 20\%$) of the observed trends at most locations open to the sea and has little impact on M_2 at the basin scale”.

- line 39-47 “Our simulations suggest...sea levels.”, these are results that should rather be in the following section Results and discussion, or in Abstract/Concluding remarks?

This is required according to the style and formatting guide of the journal (<https://www.nature.com/documents/commsj-phys-style-formatting-guide-accept.pdf>).

- line 49 : “Basic assessment” could be “Basic assessment of the model”, to be more precise
We use a model but don’t assess it. In any case, we prefer “Basic assessment” as a general starter to the Results section.

- line 52: “27 annual simulations” instead of “27 simulations” would be clearer
“27 simulations, each integrated for 40 model days” would be correct, but we have the pointer to the Methods section in case people want to read up such details (this is in keeping with the style guide). We have slightly changed the wording in this sentence to emphasize that each 40-day integration is relaxed to the time-mean stratification of the particular year (again see Methods).

- line 62 : “see Supplementary Fig. 1 and Fig. 1 (black squares)” perhaps rather “see black squares on Fig.1 and also Supplementary Fig. 1” (otherwise “Fig. 1 and Fig. 1” is confusing)
Corrected, thanks.

- line 70 “M2 sensitivity to stratification changes” rather “Sensitivity to stratification changes”
We have added two sentences on RMS variability of S_2 , K_1 , and O_1 to this section, so our original section heading can stand.

- line 73-74: “The picture is evidently dominated by temporal variability of the baroclinic tide, primarily reflecting the impact of stratification changes on wave propagation characteristics^{34, 35}”.
A general comment for this paragraph: would it be possible to clarify how the Fig. 2 can be interpreted as sensitivity to “baroclinic” tide (lines 73-87) and “barotropic” tide (lines 88-102), whereas both are mixed on Fig. 2? We guess that when strong M2 variability occur in regions of generation of internal tides (areas of rough topography, black squared on Fig. 1), it is the signature of baroclinic tide. To highlight this, black dots (similarly to black squares on Fig. 1) could be added on Fig. 2. Elsewhere, we understand that M2 variability is mainly the signature of M2 barotropic tide, is that right?

As stated on line 73, for the initial discussion of Figure 2 (lines 73–91) we do not separate barotropic from baroclinic tides, so the colored contours show their combined effect (which is nevertheless dominated by the baroclinic component). This is a natural choice for a first depiction of the signal content in the simulations. On line 92, we switch to a discussion of the barotropic tide alone. For clarity, we have added the necessary information in parentheses “... sensitivity to stratification changes in the barotropic M_2 (now separated from the baroclinic tide, see Methods).”

- line 103: “Long-term M2 trends – global synthesis” rather than “Secular M2 trends - global synthesis”
We will keep our initial section heading.

- line 106: “2D shallow-water model”, the sensitivity for the stratification is conducted with a 3D model, and sea level rise with a 2D shallow-water model, which is not fully consistent. Moreover, using the 2D shallow-water model to get rid of the sea level effect for the 3D stratification model also add complexity in the analysis (see below comment lines 333-344). We suggest to run the 3D model to investigate the sea level rise effect (or demonstrate with 1 year of simulation, that the results are similar). All the modeling part could be conducted with the 3D model and two experiments 1) an annually varying density structures simulation (correcting carefully outputs from

the steric effect, if needed), as in the paper, and 2) an annually changing bathymetry simulation, to reproduce 27 years of sea level rise.

Thanks for these comments. For the sea level rise effect, which is not the main focus of the study, we think it is the right approach to resort to a proven and simpler barotropic model. Many sea level change papers, which typically combine estimates from various sources, would not be allowed to stand if one follows through your call for 100% consistency. Moreover, 27 (or 28) simulations with the 3D model cost ~1.0 Mio core-h – a disproportional amount of electricity for a revisit of a well-studied effect.

But you are right that using the 2D model to remove (approximately) the residual sea level/steric expansion effect on M_2 was not such a good idea. As explained at the start of this letter (and the revised “Methods” section), we are now using the 3D model itself for the correction.

- Fig. 3 (a): the standard errors are quite large for M_2 trends (~ 0.05 mm/yr, Supplementary Fig. 4). Is it possible to plot only the significant M_2 trends on Fig. 3 (a)? (at 95% for example, not significant trends being for example in white, or hatched).

Done using the newly created smoothed altimetry map (with stippling, as hatching is difficult to realize with the software we are using).

- Fig. 3 (b) and (c): same comment, are all the trends significant considering the autocorrelation in the data? (See the statistical significance of trends in Methods). If not, only significant trends at 95% (for example) should be plotted (non significant trends being for example in white, or hatched). Addressed (with additional barotropic simulations and error quantification, see also our responses to Reviewer #2).

- Fig. 3: in the following of the paper, (b) and (c) are added to be compared to tide gauges trends. The same way, to be consistent, could (b) and (c) be added to be compared with (a)?

As we have written (lines 187) “we refrain from adding the altimetry solution to the comparison”. This is a deliberate choice, since more work (well beyond the present paper) is required to decipher signal from systematic error in the altimetric trends near the coast. Please see our discussion of possible issues when extracting tidal trends from satellite altimetry observations (Methods section).

- lines 126-127: “Fig. 3 shows that there are indeed contemporary large-scale trends of M_2 in the open ocean and that these signals are primarily caused by changes in ocean stratification rather than by relative sea level rise”, to demonstrate it quantitatively rather than qualitatively, would it be possible to conduct some additional diagnostics: for example, in %, how much of the observed trends (a) are captured by (b) and (c)?

Thanks, we agree that some numbers are needed here. We now include regionally averaged trends in the newly created Table 1 and discuss them (very briefly) on lines 132–135. The 3D model captures around 60% of the trends in altimetry. We omit open-ocean trends due to sea level rise from the discussion (and Table 1), since (i) they are a magnitude smaller than stratification effects, and (ii) not a focal point in this work.

- line 128: “The general tendency toward weaker M_2 amplitudes”, again, would it be possible to be quantitative? What is the global mean M_2 trend, in Fig. 3 (a), (b) and (c)?

See essentially the previous comment and response. For tidal phenomena, regional averages (such as in Table 1 or Figure 7) are a preferred means of representation.

- lines 155-157: “we again find reasonable agreement between M2 amplitude trends in the baroclinic model and those inferred from satellite altimetry” again, would it be more quantitative, with for example a global correlation map between M2 modeled time series and M2 observed times series over 1993-2019? (whose trends are Fig. 3b and 3a) Or showing what is the % of (b) compared to (a)? (i.e. how much of the observed trends are captured by the model). For example, for the “Northwest European Shelf” where there is a “reasonable agreement” (line 157), the observed trends seems very small (~ 0.05 mm/yr), compared to the modeled ones (~ 0.15 mm/yr). The % of captured trends would really help to see where the stratification process is a dominant driver.

As described in the Methods (“Satellite altimetry”) section, the altimetric solution consists of the time-mean in-phase/quadrature components and their linear trends. It is not possible to derive clean annual M_2 estimates from altimetry data; see, e.g., Ray and Egbert 2017, <https://www.taylorfrancis.com/chapters/edit/10.1201/9781315151779-13/tides-satellite-altimetry-richard-ray-gary-egbert>, for an overview of the technique. The suggestion for a correlation map is therefore off track. Concerning the other suggestion of showing “% of captured trends” in a few regions, we think that the quantifications in the current version (Table 1, Figure 7) do exactly this and are sufficient. The lines you are referring to are deliberately suggestive and are meant to support the overall message of the paragraph (trends are negative both in altimetry and the baroclinic model, hence the underlying process is an increase in conversion, not a change in vertical mixing as suggested e.g., in Müller 2012).

- lines 165-166: “We now shift the focus toward the coast and compare our simulation results against measured M2 amplitude trends at tide gauges.” As trends are probably not linear, is it possible to have at least a comparison of yearly time series at some tide gauges? (for example, in Supplementary)

The newly added Supplementary Figure 11 shows a comparison at four tide gauges (two of them are referred to in the main text). The success with which our simulations capture the year-to-year variability generally varies from location to location, and we are planning to write a follow-up paper on the subject (interannual variability without 28-year trend signal).

- line 176: “By and large, tide gauges and model results point to the same regional patterns of coastal M2 trends since 1993”, again, would it be possible to be more quantitative? Giving for example a plot ‘modeled/observed trends’, or at least the correlation by region between modeled and observed trends. It seems that some regions are better modeled than others, and large discrepancies appear for example in Fig. 5b. This could be underlined, opening the discussion to possible other drivers.

The sentence—which has been reworded for other reasons—was meant as a teaser for the subsequent discussion, which spans several paragraphs and offers many quantifications. For the plot modeled vs. observed trends (and additional quantifications), see Figure 7. Other than that:

- Correlation plots would emphasize year-to-year variability and blur any message about background trends.
- Concerning the broader discussion about possible other drivers, see our stance in our response to your general comment.
- As stated in the very beginning, discrepancies are fine and to be expected. We have included a few thoughts on model representation errors in the “Regional foci” section and under “Limitations of the 3D model” (Methods).

- line 181: “we form a “budget” of contemporary M2 amplitude trends in 13 selected coastal regions”. As shown on Fig.5, some tide gauges are well modeled, whereas others are not. We can suspect some local effects (e.g. coastal development) in tide gauges not correctly modeled. For this reason, the comparison by tide gauge may be more pertinent, than a regional budget.

There are certain limits to the spatial scales one can bridge in a self-contained paper. Here we cover global characteristics, basin wide patterns of change, aspects of tidal conversion, insights from altimetry, and spatially coherent signals along a number of coastal regions/shelf seas (Figures 5–7). That is our scope. It is impossible to add detailed comparisons for individual tide gauges, which must be left for regionally focused studies. We think this is one achievement of our work – now people know better where such studies could be worthwhile.

- line 214: “several, possibly competing processes might be at work”, yes, this may be developed more deeply in the paper. In the introduction, it is mentioned that “However, at most locations, especially those open to the sea, sea level rise alone is insufficient to explain the observed trends in tides”. The same way, in the coastal waters, Fig. 7 shows the dominant role of stratification in some regions, but in many regions (at least 5 among the 13, where less than 50% of M2 trend is captured), stratification and sea level rise can not explain alone the M2 trends.

The phrase “several, possibly competing processes ...” refers to the preceding discussion about three likely important effects in the German Bight: (a) sea level rise, (b) sediment accumulation = also a consequence of sea level rise, and (c) stratification changes. We fail to see why this is taken as an opportunity to call again for a deeper discussion (essentially everywhere). The study is framed as we think is appropriate. There is enough material, enough food for thought, and enough to follow up in future studies.

- line 219: “the primary mechanism”, as a secondary mechanism is not further discussed, we suggest the “main mechanism”?

Corrected.

- line 237: “which presently ignore changes in tides or consider them as a function of sea level rise alone”, please reformulate. The changes in tide are not always ignored. As soon as the projections are conducted with a 3D model taking into account the stratification, stratification-induced changes in tide will also be considered. If projections are based on a barotropic model, changes in tide due to stratification will effectively be ignored (but those due to mean sea level will be considered). Again, changes in tide (~0.1 mm/yr) are second order compared to changes in mean sea level (~3 mm/yr).

Exactly, projections should be conducted with a 3D model that considers the impact of stratification on tides, but no such effort exists in the literature. Moreover, one of the two reference we cite assumes tides being constant throughout the 21st century (Jevrejeva et al. 2023), the other study considers tidal changes as a function of sea level rise alone (Vousdoukas et al. 2018). Therefore, the half sentence accurately describes the current state of affairs and doesn't need to be reformulated. Concerning magnitudes, see our response to your comment starting “- line 21”.

- lines 239-277: In the “Modeling approach”, the forcing should be more explicitly mentioned. We understand that there is only the tidal forcing (earth tidal potential?) with 4 components (M2, S2, K1, O1) and associated SAL tide, and no atmospheric forcing at all (no winds and atmospheric pressure). Could this be expressed more explicitly? In particular, the line 255 is confusing “The

only forcing applied is barometric pressure loading...”. We guess this is done externally for SAL tide, and not for the simulations (otherwise, M2 trends could be also due to atmospheric forcing). Clarified with this slightly tweaked version: “... atmospheric forcing is omitted. The only forcing applied is the equilibrium tidal forcing of four primary constituents (M_2 , S_2 , K_1 , O_1 , with solid Earth tide correction) and the corresponding self-attraction and loading (SAL) tide.” The pressure loading bit referred to how the forcing is actually applied in the model but this is technicality that does not need to be mentioned.

- line 326: “26 years”, 27 years?

- line 331: “26 years”, 27 years?

We had 27 (now 28) MITgcm simulations, bounded by the mid-points of 1993 and 2019 (now 2020), so basing our auxiliary model runs on 26 (now 27) years of sea level rise is correct. In any case, the lines in question have been removed during our revisions.

- lines 333-344: this paragraph on the sea level effect linked to changes in stratification is essential and should be discussed earlier in the paper. Otherwise, from the beginning of the paper, it is not clear how to compare “stratification” and “sea level rise” whereas changes in density profiles imply changes in sea level. Concerning this paragraph, we have two major comments:

We cannot change this; the journal’s styleguide requires detailed methods to be in the Methods section. If you check out other papers from the Nature Portfolio journal, they all start with a brief (often 2–3 sentences) overview method before presenting the results. We proceed similarly but clearly can spend only half a sentence on secondary issues such as the M2/sea level correction for the MITgcm results.

[1] The model being under the Boussinesq approximation, the volumes are conserved. With this hypothesis, how the steric effect could be modeled? It is mentioned line 335: “Differences in η_0 from run to run generally reflect varying amount of steric expansion”. Is the steric effect derived a posteriori from the model temperature (Griffies and Greatbatch, 2012)? This aspect should be clarified.

Your concern and the Greatbatch correction for volume-conserving models pertain to the global mean steric effect. But what we actually refer to in this section are the local, or rather the spatially variable steric components (Note that these components are credible even when derived from Boussinesq models, see, e.g., Section 2 in Meyssignac et al., 2017, doi:10.1175/JCLI-D-17-0112.1). In our analysis, we diagnose whatever sea level/water column thickness trends are evident in the MITgcm simulations (Supplementary Figure 3a) and infer the residual M_2 trend signal from it, see the Supplementary Figure 3b. We have slightly modified the first sentence in this paragraph (→ “local trends”) to avoid readers being misled into thinking about the global mean steric effect. With the trend numbers for model sea level given in this paragraph and Supplementary Figure 3, there should be really no ambiguity of what this is about.

[2] To disentangle the steric effect from the stratification effect itself, why not simply consider the water levels (from the 3D model) corrected from the steric effect, and analyse directly M2 trends of these steric-corrected water levels? It is more direct than correcting the total M2 trends (from the 3D model), from the “residual trend signal”, computed using a barotropic model with a perturbed bathymetry to take into account the steric expansion (roughly approximated with trends from the 3D model sea levels).

Mmhhh ... sorry, we think this makes no sense. Both the change of the water column thicknesses and the resulting perturbation of M_2 occur while the model is running. So that perturbation of M_2 is in the diagnostic output, irrespective of whether or not some sort of steric correction is applied to the simulated mean sea level time series at a particular location. No changes made.

- line 370: considering the quite large standard deviation for the satellite (0.05 mm/yr), are all the trends from Fig. 3 (a) significant at 95%? If not, only significant trends should be plotted (white or hatched areas for not significant trends).

Addressed.

- line 390: GESLA-3 suggests to cite also Caldwell et al. (2015) and Woodworth et al. (2017), in addition to Haigh et al. (2022), see <https://gesla787883612.wordpress.com/downloads/>

Thanks, added.

- lines 413-421: again, are all the trends from Fig. 3b and Fig. 3c statistically significant at 95%? If not, only significant trends should be plotted (white or hatched areas).

Addressed.

Kind regards,

Lana Opel, Michael Schindelegger, Richard Ray

15th Mar 24

Dear Ms Opel,

Your manuscript titled "A likely role for stratification in secular changes of the global ocean tides" has now been seen by our reviewers, whose comments appear below. In light of their advice we are delighted to say that we are happy, in principle, to publish a suitably revised version in Communications Earth & Environment under the open access CC BY license (Creative Commons Attribution v4.0 International License).

We therefore invite you to revise your paper one last time to address the remaining concerns of our reviewers. In particular, please carefully consider the use of your terminology in light of reviewer 2's comments, and use "secular" only where you refer to century-scale changes.

At the same time we ask that you edit your manuscript to comply with our format requirements and to maximise the accessibility and therefore the impact of your work.

EDITORIAL REQUESTS:

*****Please take care to match our formatting and policy requirements. We will check revised manuscript and return manuscripts that do not comply. Such requests will lead to delays. *****

SUBMISSION INFORMATION:

OPEN ACCESS:

Communications Earth & Environment is a fully open access journal. Articles are made freely accessible on publication under a CC BY license (Creative Commons Attribution 4.0 International License). This license allows maximum dissemination and re-use of open access materials and is preferred by many research funding bodies.

For further information about article processing charges, open access funding, and advice and support from Nature Research, please visit <https://www.nature.com/commsenv/article-processing-charges>

At acceptance, you will be provided with instructions for completing this CC BY license on behalf of all authors. This grants us the necessary permissions to publish your paper. Additionally, you will be asked to declare that all required third party permissions have been obtained, and to provide billing information in order to pay the article-processing charge (APC).

[link redacted]

Best regards,

Jose Luis Iriarte Machuca, PhD

Editorial Board Member

Communications Earth & Environment

Heike Langenberg, PhD

Chief Editor

Communications Earth & Environment

On X(Twitter): @CommsEarth

REVIEWERS' COMMENTS:

Reviewer #2 (Remarks to the Author):

I would like to thank the authors for the rigor of their answers to the questions and remarks, the reprocess and addition of figures as well as the additional runs of the 2D model. The paper is now a lot more clearer to me – especially the figure 3 which is the key one of the paper – and I do not doubt that it will help to further address the question of stratification in tidal changes.

Authors propose a the mention “(S. Barbot, personal communication, 2023)” to insert a remark on the nudging method used in the 3D model (Methods > Modeling approach). I am not very familiar with such convention, but as long as it seems right to the editors, I agree to this mention.

Best regards,

Simon Barbot

Last minor comment :

Supplement Fig 11: You did not show the stipplings of trend significance as in the other figures.

Reviewer #3 (Remarks to the Author):

We thank the authors for the revised manuscript, and their response to our comments. We found that Fig. 3 has really been improved, and it is a good point to see that one of the recommendation ('consistency issue' for the steric expansion effect, see your second point) was useful. The paper is a nice contribution to the question of long-term tidal changes. However, we are not fully in line with some of the responses, and here are some additional comments, we would like to share with you. The line numbers are those of previous review.

- line 1 : we still believe that “secular changes” is appropriate when referring to a century and not a short period of 28 years (1993-2020). Astronomers mainly used the term “secular” in this context, the word coming from the Latin word ‘saeculum’ which refers to a century’s duration.

Accordingly, many reference papers referring to “secular” changes analyze data over a century. For example, Ray (2006) in “Secular changes of the M2 tide in the Gulf of Maine” analyzed data over 1905-2004 at St John, in the Gulf of Maine. Ray (2009) in “Secular changes in the solar semidiurnal tide of the western North Atlantic Ocean” analyzed data over 1910-2010. Woodworth et al. (1991) in “Secular trends in mean tidal range around the British Isles and along the adjacent European coastline” analyzed tide gauges spanning from 1862 to 1988, and Cartwright (1972) in “Secular changes in the oceanic tides at Brest” analyzed data from 1711 to the mid-20th century. We believe that trends on periods as short as 28 years should not be referred as “secular trends”, which may be confusing. The confusion may be even amplified, when trends computed on short altimetry periods (around 30 years) are extrapolated on the century, see for example the paper you mention, Bij de Vaate et al. (2022) for the altimetry time span. They report that “The amplitudes of the considered tides have changed by up to 1 mm/year over the past ~3 decades. This implies a change of up to 10 cm per century.” Such a large change is based on the hypothesis of linear changes, which is not always verified, see for example Pouvreau et al. (2006) at Brest, Müller (2011) in the North Atlantic, Ray and Talke (2019) in the Gulf of Maine. Note that a century ago, with only 22 years of data at St John, Doodson (1924) “surmised a secular change in M2, but his time series turned out too short (22 years) to determine definitively even the sign of this change.” (Ray, 2006).

To conclude, we believe that “long-term” changes would be more appropriate than “secular” changes, as satellite data cover only 28 years, and knowing the strong interannual to multidecadal variability in sea level data (IPCC, 2021).

- line 8: “The quoted -0.1 mm/yr for open-ocean trends is an order of magnitude estimate, saying that the regionally coherent trends are neither -0.01 mm/yr nor -1 mm/yr. Adding uncertainty information to such an estimate makes little sense (apart from being very difficult, given that confidence intervals cover a wide range of values).” We understand that it is difficult to give confidence intervals, but giving uncertainty information on trends (when possible) makes sense.

- line 16: “the broader oceanic factors” has been replaced by the “natural factors”, rather than “the broader oceanic and atmospheric factors” as suggested. The possible link between the broader atmospheric factors and changes in tide is discussed in the literature (e.g. Muller, 2011; Pan et al., 2019; Challis et al., 2023), but it seems neglected in the present paper.

- line 21: "Changes of ~3–5 cm in M2 by 2100 are possible and that would be as large as the extreme sea level contribution due to storm surges and waves". Again, 3-5 cm by 2100 in M2 is completely second order compared to changes in mean sea level (at least 10 times greater, 28-55 cm under the very low GHG emissions scenario, IPCC, 2021). My previous comment was that changes in extreme sea levels (=MSL+tide+storm surges, wave setup being part of storm surges) are firstly driven by sea level rise, and changes in tide are from far second order. Of course, the subject is worth investigating.

- lines 73-74: "the contours show their combined effect (which is nevertheless dominated by the baroclinic component)". Again, regions of generation of internal tides could be added to highlight this. Outside of these regions, I guess that M2 variability is mainly due to barotropic component?

- lines 333-344 point [2] The suggestion of analyzing directly steric-corrected sea levels from your 3D model makes sense. Here, you compute M2 on the total water levels, and then remove (approximately) the residual sea level/steric expansion effect on M2, removing a residual trend (estimated with a model). I was just wondering why not correcting your 3D total water levels from the steric expansion (in the post-processing), and then analyzing directly M2 on these steric-corrected sea levels (this approach would avoid applying your approximated trend correction).

Dear Editors and Reviewers,

We thank you for evaluating the revised manuscript and the constructive feedback. Below are our point-by-point responses to the remaining comments and questions.

(Comments received on 15 March 2024)

Reviewer #2:

I would like to thank the authors for the rigor of their answers to the questions and remarks, the reprocess and addition of figures as well as the additional runs of the 2D model. The paper is now a lot clearer to me – especially the figure 3 which is the key one of the paper – and I do not doubt that it will help to further address the question of stratification in tidal changes. Authors propose to mention “(S. Barbot, personal communication, 2023)” to insert a remark on the nudging method used in the 3D model (Methods > Modeling approach). I am not very familiar with such convention, but as long as it seems right to the editors, I agree to this mention.

Best regards,
Simon Barbot

Last minor comment:

Supplement Fig 11: You did not show the stipplings of trend significance as in the other figures. Stippling now added. Thank you for all the good suggestions and keen observations.

Reviewer #3:

We thank the authors for the revised manuscript, and their response to our comments. We found that Fig. 3 has really been improved, and it is a good point to see that one of the recommendations (‘consistency issue’ for the steric expansion effect, see your second point) was useful. The paper is a nice contribution to the question of long-term tidal changes. However, we are not fully in line with some of the responses, and here are some additional comments, we would like to share with you. The line numbers are those of previous review.

- line 1: we still believe that “secular changes” is appropriate when referring to a century and not a short period of 28 years (1993-2020). Astronomers mainly used the term “secular” in this context, the word coming from the Latin word ‘saeculum’ which refers to a century’s duration.

Accordingly, many reference papers referring to “secular” changes analyze data over a century. For example, Ray (2006) in “Secular changes of the M2 tide in the Gulf of Maine” analyzed data over 1905-2004 at St John, in the Gulf of Maine. Ray (2009) in “Secular changes in the solar semidiurnal tide of the western North Atlantic Ocean” analyzed data over 1910-2010. Woodworth et al. (1991) in

“Secular trends in mean tidal range around the British Isles and along the adjacent European coastline” analyzed tide gauges spanning from 1862 to 1988, and Cartwright (1972) in “Secular changes in the oceanic tides at Brest” analyzed data from 1711 to the mid-20th century. We believe that trends on periods as short as 28 years should not be referred as “secular trends”, which may be confusing. The confusion may be even amplified, when trends computed on short altimetry periods (around 30 years) are extrapolated on the century, see for example the paper you mention, Bij de Vaate et al. (2022) for the altimetry time span. They report that “The amplitudes of the considered tides have changed by up to 1 mm/year over the past ~3 decades. This implies a change of up to 10 cm per century.” Such a large change is based on the hypothesis of linear changes, which is not always verified, see for example Pouvreau et al. (2006) at Brest, Müller (2011) in the North Atlantic, Ray and Talke (2019) in the Gulf of Maine. Note that a century ago, with only 22 years of data at St John, Doodson (1924) “surmised a secular change in M₂, but his time series turned out too short (22 years) to determine definitively even the sign of this change.” (Ray, 2006).

To conclude, we believe that “long-term” changes would be more appropriate than “secular” changes, as satellite data cover only 28 years, and knowing the strong interannual to multidecadal variability in sea level data (IPCC, 2021).

Thank you for your elaborations and pointing out the origin of the word “secular”. Accordingly, we have replaced the word “secular” with “long-term” in the title and similarly in the main text (sometimes also writing “contemporary trends” or just “trends” instead of “secular trends”). That these trends hold for the 1993–2020 time period is clear from the respective passages. Note that on line 46 of the revised manuscript, we still write “The study adds new facets to the discussion of secular changes in tides ...”, emphasizing that stratification changes are now a factor to consider when studying century-long trends.

- line 8: “The quoted -0.1 mm/yr for open-ocean trends is an order of magnitude estimate, saying that the regionally coherent trends are neither -0.01 mm/yr nor -1 mm/yr. Adding uncertainty information to such an estimate makes little sense (apart from being very difficult, given that confidence intervals cover a wide range of values).” We understand that it is difficult to give confidence intervals, but giving uncertainty information on trends (when possible) makes sense.

We maintain our ground that the -0.1 mm/yr magnitude estimate in the abstract does not require uncertainty information. We nevertheless perused the text once more and added confidence levels for three altimetry trend estimates in Section “M₂ trends—global synthesis” (lines 129, 149, 159).

- line 16: “the broader oceanic factors” has been replaced by the “natural factors”, rather than “the broader oceanic and atmospheric factors” as suggested. The possible link between the broader atmospheric factors and changes in tide is discussed in the literature (e.g. Muller, 2011; Pan et al., 2019; Challis et al., 2023), but it seems neglected in the present paper.

The evidence for a link of tidal changes to atmospheric factors (related to, e.g., NAO, Müller 2011, Pan et al. 2019) is fairly sketchy. In fact, the temporal changes of the M₂ nodal cycle in the Gulf of Maine, as analyzed in Pan et al. (2019), appear to be manifestations of strong interannual M₂ variability, again driven by stratification changes near steep topography (see “Conclusions” and SI of Schindelegger et al. 2022, <https://doi.org/10.1029/2022GL101671>). As for Müller (2011), we have been able to reproduce part of the basin-wide M₂ decrease during the 1980s in the North Atlantic, using

the very same modeling strategy as in the present work, just with decadal time slices and WOA2018 hydrography. The underlying shift in stratification may indeed be due to NAO, but this is quite speculative. The only more cogent study on atmospheric influences on (regional) tides is Challis et al. (2023). We have added a citation to their paper on line 11, that is, the first sentence of the Introduction, which was generally aimed to highlight relevant works by a diverse mix of authors, focusing on different regions, time scales, and processes.

- line 21: “Changes of ~3–5 cm in M2 by 2100 are possible and that would be as large as the extreme sea level contribution due to storm surges and waves”. Again, 3-5 cm by 2100 in M2 is completely second order compared to changes in mean sea level (at least 10 times greater, 28-55 cm under the very low GHG emissions scenario, IPCC, 2021). My previous comment was that changes in extreme sea levels (=MSL+tide+storm surges, wave setup being part of storm surges) are firstly driven by sea level rise, and changes in tide are from far second order. Of course, the subject is worth investigating.

We have already toned down the sentences in question in the previous revision, to avoid overemphasis on the tidal contribution to extreme sea levels. In any case, we do not attempt a comparison of magnitudes of different drivers, but rather highlight physical processes that are yet to be included in rigorous projections (and surely, physically complete models are desirable; that’s something we would not want to tone down).

- lines 73-74: “the [colored] contours show their combined effect (which is nevertheless dominated by the baroclinic component)”. Again, regions of generation of internal tides could be added to highlight this. Outside of these regions, I guess that M2 variability is mainly due to barotropic component?

The regions of internal tide generation are highlighted in Fig. 1, which is discussed a few sentences before (lines 60–64), so the context should be clear. Adding them again to Fig. 2 would make the plot very cluttered. Variability in the barotropic component is highlighted by the thick dotted contour in Fig. 2, as noted in the caption. No changes made.

- lines 333-344 point [2] The suggestion of analyzing directly steric-corrected sea levels from your 3D model makes sense. Here, you compute M2 on the total water levels, and then remove (approximatively) the residual sea level/steric expansion effect on M2, removing a residual trend (estimated with a model). I was just wondering why not correcting your 3D total water levels from the steric expansion (in the post-processing), and then analyzing directly M2 on these steric-corrected sea levels (this approach would avoid applying your approximated trend correction).

OK, thanks for clarifying your reasoning. We would still argue that the suggested approach doesn’t solve the problem in post-processing. For tides, which are inherently linked to oceanic normal modes and resonance, water column thickness changes at one location (e.g., due to steric expansion in the model) can affect tidal properties elsewhere. This is in fact what we are seeing in Fig. 3 in the North Atlantic. Our trend correction, while approximate, is the most reasonable approach to pursue here, but good that you have prompted us to switch from the barotropic model to the 3D model for that particular step.

Kind regards,

Lana Opel, Michael Schindelegger, Richard Ray